# INPP4B promotes PI3Kα-dependent late endosome formation and Wnt/β-catenin signaling in breast cancer

Samuel J. Rodgers[1,2], Lisa M. Ooms [1,2], Viola M. J. Oorschot[3,10], Ralf B. Schittenhelm [4], Elizabeth V. Nguyen[1,2], Sabryn A. Hamila[1,2], Natalie Rynkiewicz[2,11], Rajendra Gurung[1,2], Matthew J. Eramo[1,2], Absorn Sriratana[1,2], Clare G. Fedele[2,12], Franco Caramia [5], Sherene Loi [5], Genevieve Kerr[6,7], Helen E. Abud [6,7], Georg Ramm [1,2,3], Antonella Papa [1,2], Andrew M. Ellisdon[1,2,8], Roger J. Daly [1,2], Catriona A. McLean [9] & Christina A. Mitchell [1,2✉]

INPP4B suppresses PI3K/AKT signaling by converting PI(3,4)P$_2$ to PI(3)P and INPP4B inactivation is common in triple-negative breast cancer. Paradoxically, INPP4B is also a reported oncogene in other cancers. How these opposing INPP4B roles relate to PI3K regulation is unclear. We report *PIK3CA*-mutant ER$^+$ breast cancers exhibit increased INPP4B mRNA and protein expression and INPP4B increased the proliferation and tumor growth of *PIK3CA*-mutant ER$^+$ breast cancer cells, despite suppression of AKT signaling. We used integrated proteomics, transcriptomics and imaging to demonstrate INPP4B localized to late endosomes via interaction with Rab7, which increased endosomal PI3Kα-dependent PI(3,4) P$_2$ to PI(3)P conversion, late endosome/lysosome number and cargo trafficking, resulting in enhanced GSK3β lysosomal degradation and activation of Wnt/β-catenin signaling. Mechanistically, Wnt inhibition or depletion of the PI(3)P-effector, Hrs, reduced INPP4B-mediated cell proliferation and tumor growth. Therefore, INPP4B facilitates PI3Kα crosstalk with Wnt signaling in ER$^+$ breast cancer via PI(3,4)P$_2$ to PI(3)P conversion on late endosomes, suggesting these tumors may be targeted with combined PI3K and Wnt/β-catenin therapies.

[1] Cancer Program, Biomedicine Discovery Institute, Monash University, Clayton, VIC, Australia. [2] Department of Biochemistry and Molecular Biology, Monash University, Clayton, VIC, Australia. [3] Monash Ramaciotti Centre for Cryo Electron Microscopy, a Node of Microscopy Australia, Monash University, Victoria, Australia. [4] Monash Proteomics and Metabolomics Facility, Monash Biomedicine Discovery Institute and Department of Biochemistry and Molecular Biology, Monash University, Clayton, VIC, Australia. [5] Peter MacCallum Cancer Centre, University of Melbourne, Melbourne, VIC, Australia. [6] Development and Stem Cells Program, Biomedicine Discovery Institute, Monash University, Clayton, VIC, Australia. [7] Department of Anatomy and Developmental Biology, Monash University, Clayton, VIC, Australia. [8] Australian Research Council Centre of Excellence in Advanced Molecular Imaging, Monash University, Clayton, VIC, Australia. [9] Department of Anatomical Pathology, Alfred Hospital, Prahran, VIC, Australia. [10] Present address: Electron Microscopy Core Facility, European Molecular Biology Laboratory, Heidelberg, Germany. [11] Present address: Babraham Institute, Cambridge, UK. [12] Present address: Peter MacCallum Cancer Centre, University of Melbourne, Melbourne, VIC, Australia. ✉email: christina.mitchell@monash.edu

The phosphoinositide 3-kinase (PI3K) signaling pathway is frequently hyperactivated in cancer. Class I PI3K transiently generates phosphatidylinositol 3,4,5-trisphosphate $(PI(3,4,5)P_3)$ on the inner leaflet of the plasma membrane in response to extracellular stimulation, which is rapidly converted to phosphatidylinositol 3,4-bisphosphate $(PI(3,4)P_2)$ by inositol polyphosphate 5-phosphatases[1]. $PI(3,4,5)P_3$ and $PI(3,4)P_2$ together increase the recruitment and activation of the protein kinase AKT[2,3], the central effector of oncogenic PI3K signaling. $PI(3,4,5)P_3/PI(3,4)P_2$ signaling activates numerous other effectors at the plasma membrane, including the protein kinase PDK1, which in turn phosphorylates AKT and other AGC kinases such as SGK3[4]. The tumor suppressor, phosphatase and tensin homolog (PTEN), converts $PI(3,4,5)P_3$ to $PI(4,5)P_2$ thereby inhibiting PI3K/AKT signaling. Inositol polyphosphate 4-phosphatase type II (INPP4B) dephosphorylates $PI(3,4)P_2$ downstream of class I PI3K to form phosphatidylinositol 3-phosphate (PI(3)P), also suppresses AKT signaling, and is proposed to function as a tumor suppressor in some cancers[5,6].

$PIK3CA$, which encodes the p110α subunit of PI3Kα, is mutated in up to 40% of breast cancers, most frequently in estrogen receptor-positive $(ER^+)$ tumors[7,8]. Mutant $PIK3CA$ is not an independent marker of prognosis when corrected for favorable-risk variables such as ER-positivity[9,10], but is a predictive biomarker for improved response to combined endocrine and PI3K therapy (alpelisib-fulvestrant) in advanced $ER^+$ breast cancers[11]. Interestingly, $PIK3CA$-mutant $ER^+$ breast cancers display minimal AKT activation and little dependence on AKT signaling compared to tumors with other PI3K pathway alterations, such as $PTEN$ loss[4,12].

We and others have identified that INPP4B inhibits AKT signaling and exhibits tumor suppressor activity in triple-negative $(ER^-/PR^-/HER2^-)$ and basal-like breast cancers[5,6,13]. In addition, many subsequent reports have shown that INPP4B protein expression is reduced in melanoma, ovarian, and prostate cancers[5,14,15]. In murine models, $Inpp4b$ ablation increases mammary tumor penetrance in cooperation with $Tp53/Brca1$ deletion[13], and promotes thyroid tumorigenesis and metastasis in vivo in cooperation with $Pten$ heterozygous loss[16,17]. Notably, INPP4B suppresses localized AKT2 signaling on early endosomes and EGFR degradation by degrading $PI(3,4)P_2$[13,16,18].

Paradoxically, more recent studies have indicated a possible oncogenic role for INPP4B in acute myeloid leukemia (AML) associated with chemoresistance, as well as in melanoma and colon cancer[19–22]. Several molecular mechanisms for INPP4B oncogenic function have been proposed via regulation of PTEN protein stability[21], or SGK3 activation[22,23], and may be context-dependent. For example, in breast cancer SGK3 is amplified and its kinase activity is dependent on oncogenic PI3K and INPP4B[23]. Recently INPP4B was identified as the top gene associated with $ER^+$ breast cancers and tumor grade[24].

Here, we further explored the role INPP4B plays in $ER^+$ breast cancer revealing a cohort of $PIK3CA$-mutant $ER^+$ breast cancers exhibit increased INPP4B expression. We demonstrate INPP4B overexpression enhances the proliferation and tumor growth of $PIK3CA$-mutant $ER^+$ breast cancer cell lines, despite concurrent AKT signaling suppression. To investigate this apparent paradox, we utilized an integrative approach featuring proteomics, transcriptomics, and advanced cellular imaging to elucidate the molecular mechanisms by which INPP4B promotes tumorigenesis of $PIK3CA$-mutant $ER^+$ breast cancers. Our findings show INPP4B stimulates a PI3Kα-dependent signaling hub on late endosomes that directs Wnt/β-catenin activation. These studies reveal a mechanism for crosstalk between two oncogenic signaling pathways that promotes cell proliferation and tumor growth.

## Results

**INPP4B expression is increased in *PIK3CA*-mutant ER⁺ breast cancer.** Recently INPP4B was identified as a bona fide marker for ER-positivity in human breast cancer[24]. INPP4B expression is lost in triple-negative breast cancers, however, its emergence as a potential oncogene in other cancers led us to examine its relative expression and function in $ER^+$ breast cancers. Here, INPP4B protein expression was examined by immunohistochemistry using a validated monoclonal INPP4B (3D5) antibody[6] on two independent tissue cohorts; one tissue microarray containing 224 primary human breast cancers with 32 tumor-adjacent normal breast tissues (US Biomax) (Supplementary Fig. 1a), and a second cohort of 107 primary breast cancers collected as part of the Melbourne Collaborative Cohort Study (MCCS)[6]. Loss of INPP4B protein expression was associated with triple-negative breast cancers (Fig. 1a, b), as we previously reported[6]. However, an increase in INPP4B protein expression was observed in 14–40% of breast cancers relative to normal tissue associated with ER/PR-positivity and the luminal ($ER^+$ and/or $PR^+$) breast cancer subtype (Fig. 1a, b and Supplementary Fig. 1b, c). As mRNA expression data were not available for these cohorts, $INPP4B$ mRNA expression was examined using Tissue Scan Breast Cancer cDNA arrays I–IV (OriGene) of 130 primary human breast cancers and 16 normal breast tissues (Fig. 1c). Decreased $INPP4B$ expression was associated with triple-negative breast cancers, and notably increased $INPP4B$ expression was demonstrated in 25% of breast cancers associated with ER/PR-positivity and the luminal subtype (Fig. 1c, d and Supplementary Fig. 1d, e).

Mutations in PI3K pathway genes $PIK3CA$, $PTEN$, and $AKT1$ are commonly associated with $ER^+$ breast cancers[12]. Using the METABRIC and TCGA datasets[8,25], we found that only 1% of breast cancers exhibited $INPP4B$ genetic alterations such as mutations, truncations, amplifications, or deletions. $INPP4B$ mRNA expression did not associate with $PTEN$ or $AKT1$ mutations, but positively correlated with $PIK3CA$-mutation status (Fig. 1e and Supplementary Fig. 1f). $INPP4B$ expression was also significantly higher in breast cancers with multiple $PIK3CA$ mutations (Supplementary Fig. 1g). Further stratification revealed high $INPP4B$ expression was specifically associated with the $PIK3CA$-mutant $ER^+$ breast cancer subset (Fig. 1f).

**INPP4B enhances *PIK3CA*-mutant ER⁺ breast cancer and mammary epithelial cell proliferation.** To determine the functional consequences of increased INPP4B expression in $PIK3CA$-mutant $ER^+$ breast cancer, GFP-INPP4B was expressed in MCF-7 and T47D $ER^+$ breast cancer cells (Supplementary Fig. 2a), which harbor hyperactivating $PIK3CA^{E545K}$ and $PIK3CA^{H1047R}$ mutations, respectively. The ability of cancer cells to undergo anchorage-independent cell growth is a hallmark of cellular transformation. GFP-INPP4B significantly increased colony size of MCF-7 (2.1-fold) and T47D (1.8-fold) cells in soft agar but had no effect on colony number (Fig. 2a–c). Expression of Myc-INPP4B$^{WT}$, but not a $PI(3,4)P_2$ 4-phosphatase-dead Myc-INPP4B$^{C842A}$ mutant[19] (Supplementary Fig. 2b), increased MCF-7 cell colony size (2.1-fold) in soft agar but did not affect colony number (Supplementary Fig. 2c–e), consistent with increased cell proliferation. GFP-INPP4B enhanced MCF-7 cell proliferation (1.7-fold) following serum deprivation relative to vector controls (Fig. 2d, e), but did not affect the proliferation of MCF-7 cells grown in serum-containing media (Supplementary Fig. 2f). EGF-stimulated AKT$^{S473}$ and AKT$^{T308}$ phosphorylation levels were modestly reduced in MCF-7 cells overexpressing GFP-INPP4B (Fig. 2f–h), revealing that INPP4B promotes cell

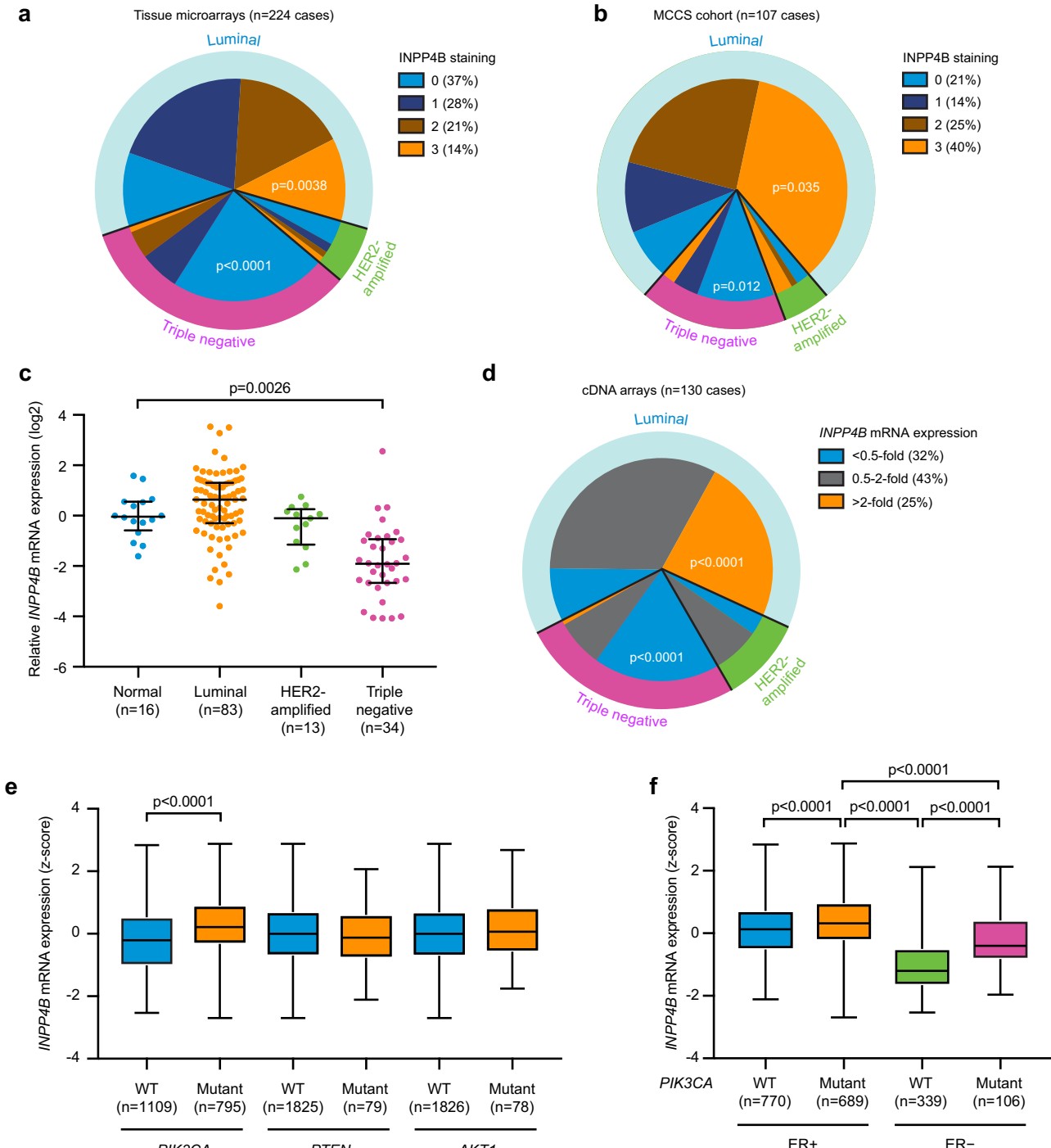

**Fig. 1 INPP4B expression is increased in *PIK3CA*-mutant ER⁺ breast cancers. a, b** Primary human breast cancer tissues and normal adjacent breast tissues from tissue microarrays (US Biomax, $n = 224$ cases) (**a**) or the Melbourne Collaborative Cohort Study (MCCS, $n = 107$ cases) (**b**) were immunostained using an INPP4B-specific monoclonal antibody. INPP4B protein expression was scored as 0 (no expression), 1 (low), 2 (moderate), and 3 (high), and correlated with breast cancer subtype. **c, d** Relative *INPP4B* mRNA expression was quantified from normal adjacent breast and primary breast cancer tissue samples using Tissue Scan Breast Cancer cDNA Arrays I–IV (OriGene) by quantitative RT-PCR using *INPP4B* primers ($n = 130$ cases). *INPP4B* mRNA expression was normalized to *β-ACTIN* expression levels and quantified relative to the mean of the normal adjacent breast samples. **c** Data represent median *INPP4B* mRNA expression ±25th and 75th percentiles. **d** Altered *INPP4B* mRNA expression was correlated with breast cancer subtype. **e, f** *INPP4B* mRNA expression was stratified by *PIK3CA*, *PTEN*, or *AKT1* mutation status (**e**), or *PIK3CA* mutation status and ER-positivity (**f**) in breast tumors from the METABRIC cohort ($n = 1904$ cases). The center line indicates the median, the lower bound of the box indicates the 25th percentile, the upper bound of the box represent the 75th percentile, the lower whisker extends from the 25th percentile to the minimum value, and the upper whisker extends from the 75th percentile to the maximum value. *p* values determined by Fisher's exact test are indicated in **a, b, d**, by Kruskal–Wallis test with Dunn's post hoc test in **c, f**, and by two-tailed unpaired Mann–Whitney test in **e**.

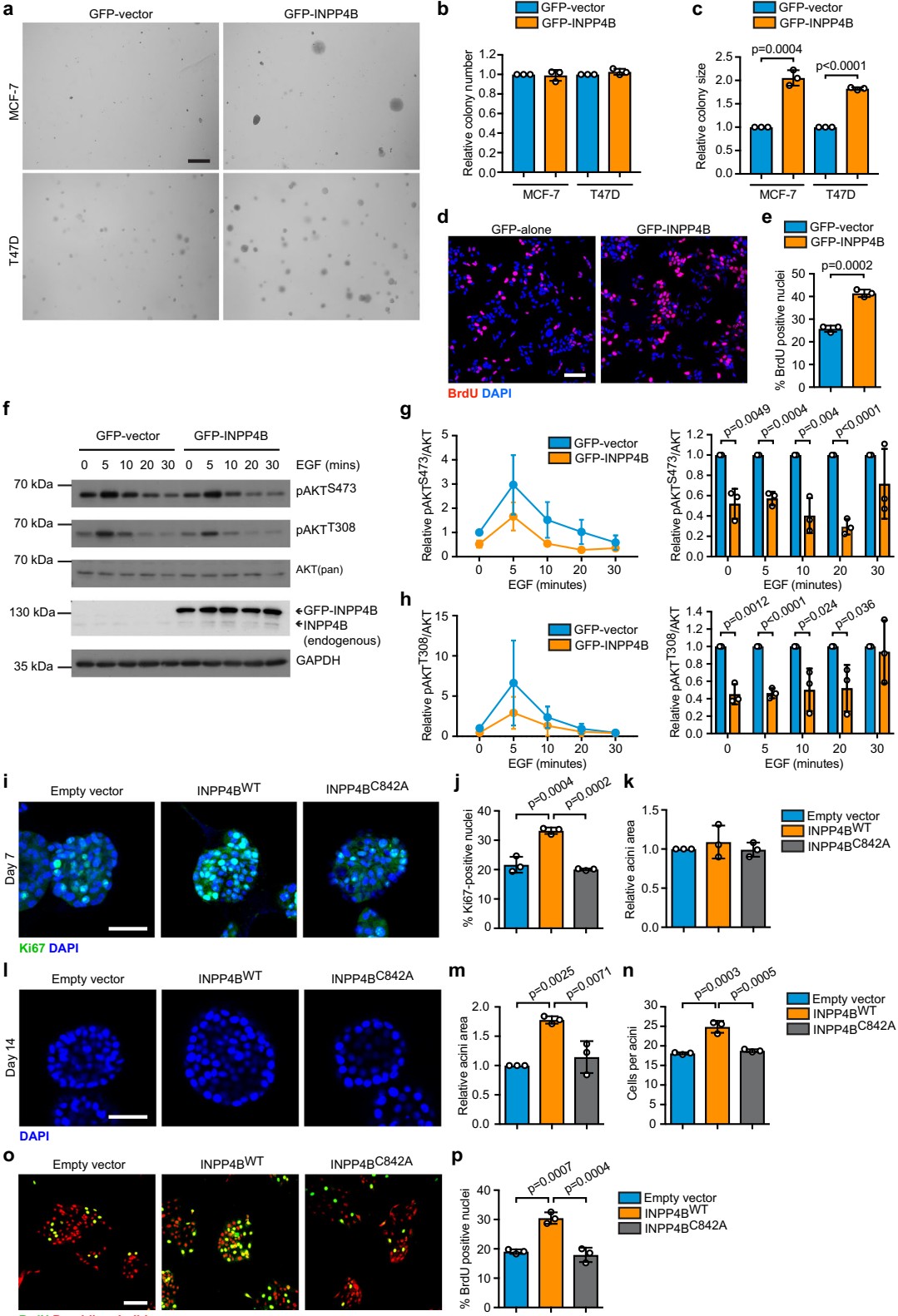

proliferation despite reducing AKT activation. In addition, GFP-INPP4B expression had no effect on PTEN protein levels (Supplementary Fig. 2g, h) or EGF-mediated SGK3$^{T320}$ phosphorylation (Supplementary Fig. 2i, j), suggesting the INPP4B phenotype we observed is not mediated by alterations in PTEN and/or SGK3 signaling. These results differ from that described by Gasser et al. who used IGF-1, rather than EGF stimulation as

used here to activate AKT and SGK3 signaling[23]. We also cannot exclude the possibility that the level of INPP4B overexpression generated by our studies was insufficient to hyperactivate SGK3.

Next, we assessed whether INPP4B regulates the morphogenesis of three-dimensional epithelial spheroids using MCF-10A mammary cells that express wild-type *PIK3CA*, which form polarized, growth-limited acini in 3D Matrigel culture[26]. After

**Fig. 2 INPP4B promotes *PIK3CA*-mutant ER⁺ breast cancer and mammary epithelial cell proliferation. a–c** MCF-7 or T47D cells expressing GFP-INPP4B or GFP-vector were suspended in 0.3% soft agar and cultured for 4 weeks to allow anchorage-independent cell growth (**a**). Data represent relative number of colonies (**b**), and relative colony size (**c**) ($n > 50$ colonies/experiment) ± SD ($n = 3$ experiments in triplicate). **d, e** MCF-7 cells expressing GFP-INPP4B or GFP-vector were serum-starved for 24 h, incubated with BrdU overnight then fixed and immunostained with BrdU antibodies and DAPI (**d**). Data represent mean percentage of BrdU-positive cells ± SD ($n = 3$ experiments, >300 cells/experiment) (**e**). **f–h** MCF-7 cells expressing GFP-INPP4B or GFP-vector were serum-starved overnight, then stimulated with EGF (100 ng/mL) for the indicated times. Cells were lysed and immunoblotted with pAKT$^{S473}$, pAKT$^{T308}$, or AKT(pan) antibodies or GAPDH antibodies as a loading control (**f**). Data represent the mean relative pAKT$^{S473}$ (**g**) or pAKT$^{T308}$ (**h**) levels relative to AKT(pan) ± SD ($n = 3$ experiments). **i–k** MCF-10A cells expressing Myc-INPP4B$^{WT}$, Myc-INPP4B$^{C842A}$, or empty vector were cultured in Matrigel for 7 days, then fixed and immunostained with Ki67 antibodies and DAPI (**i**). Data represent mean percentage of Ki67-positive cells ± SD ($n = 3$ experiments, >200 cells/experiment) (**j**), and relative acini area ± SD ($n = 3$ experiments, 15 largest acini scored/experiment) (**k**). **l–n** MCF-10A cells expressing INPP4B$^{WT}$, INPP4B$^{C842A}$, or empty vector were cultured in Matrigel for 14 days, then fixed and immunostained with DAPI (**l**). Data represent relative acini area (**m**) and number of cells per acinus (**n**) ± SD ($n = 3$ experiments, 15 largest acini scored/experiment). **o, p** MCF-10A cells expressing Myc-INPP4B$^{WT}$, Myc-INPP4B$^{C842A}$, or empty vector were serum-starved for 48 h, then incubated with BrdU overnight. Cells were fixed and immunostained with BrdU antibodies and propidium iodide (**o**). Data represent mean percentage of BrdU-positive cells ± SD ($n = 3$ experiments, >300 cells/experiment) (**p**). Scale bars, 50 μm (**i, l**), 100 μm (**d, o**), 1 mm (**a**). *p* values determined by two-tailed unpaired *t*-test are indicated in **c, e, g, h**, or by one-way ANOVA with Tukey post hoc test in **j, m, n, p**.

7 days in culture, Myc-INPP4B$^{WT}$ but not Myc-INPP4B$^{C842A}$ (PI(3,4)P$_2$ 4-phosphatase-dead) expressing MCF-10A acini (Supplementary Fig. 2k) displayed a significant increase in the percentage of proliferating cells compared to vector control acini (1.5-fold) (Fig. 2i, j). No significant changes to acini size were observed at day 7 (Fig. 2k). By day 14, Myc-INPP4B$^{WT}$ but not Myc-INPP4B$^{C842A}$ expressing MCF-10A cells formed larger acini (1.8-fold), with an increased number of cells per acinus (1.4-fold) compared to vector controls (Fig. 2l–n). After 21 days, Myc-INPP4B$^{WT}$ overexpressing MCF-10A cells formed mature acini with a single layer of epithelial cells surrounding a hollow lumen (Supplementary Fig. 2l). Myc-INPP4B$^{WT}$ expression did not affect the apical distribution of the Golgi marker GM130 (Supplementary Fig. 2m), or the percentage of cleaved caspase-3-positive acini (Supplementary Fig. 2n, o). This suggests that INPP4B overexpression promotes MCF-10A cell proliferation but does not affect apico-basal polarity nor apoptosis. Consistent with this, Myc-INPP4B$^{WT}$ but not Myc-INPP4B$^{C842A}$ overexpression enhanced MCF-10A cell proliferation in monolayer cultures following serum deprivation (1.7-fold) (Fig. 2o, p). EGF-stimulated AKT$^{S473}$ phosphorylation was decreased in Myc-INPP4B$^{WT}$ but not Myc-INPP4B$^{C842A}$ overexpressing MCF-10A cells (Supplementary Fig. 2p, q). This data is consistent with an interpretation that INPP4B promotes the proliferation of *PIK3CA*-mutant ER⁺ breast cancer and *PIK3CA*-wild-type mammary epithelial cells in a PI(3,4)P$_2$ 4-phosphatase-dependent manner.

**INPP4B promotes PI(3,4)P$_2$ to PI(3)P conversion on late endosomes**. We utilized several unbiased proteomics-based techniques to characterize the downstream signaling functions of INPP4B in *PIK3CA*-mutant ER⁺ breast cancer. First, whole cell proteomics using liquid chromatography-tandem mass spectrometry (LC-MS/MS) was undertaken to identify variations in protein levels (>2-fold) in GFP-INPP4B versus GFP-vector expressing MCF-7 cells. Functional annotation analysis revealed that the upregulated proteins in INPP4B-overexpressing cells were strongly associated with the endolysosomal system, the vesicular trafficking pathway that mediates the internalization and degradation of extracellular receptors and cargo (Fig. 3a). Furthermore, immunoprecipitation-mass spectrometry (IP-MS) analysis of GFP-INPP4B protein complexes was performed on MCF-7 cells stimulated with EGF to activate class I PI3K signaling. The most-enriched binding partner identified from this analysis was RAB7A (Rab7), a small GTPase that localizes to late endosomes (Fig. 3b). Binding between INPP4B and Rab7 was confirmed by co-immunoprecipitation of GFP-INPP4B with

endogenous Rab7 (Fig. 3c). Rab7 regulates late endosome formation, motility, and fusion with lysosomes, and its localization and function is controlled by GTP/GDP loading, whereby it localizes to late endosomes in its active GTP-bound state and then dissociates following hydrolysis of GTP to GDP[27]. Immunoprecipitation of HA-Rab7$^{WT}$, HA-Rab7$^{Q67L}$ (constitutively GTP-bound) or HA-Rab7$^{T22N}$ (constitutively GDP-bound), showed that endogenous INPP4B was able to bind both GTP and GDP-loaded Rab7 (Supplementary Fig. 3a). Notably, in a purified in vitro component assay recombinant Rab7 significantly enhanced the PI(3,4)P$_2$ 4-phosphatase activity of recombinant INPP4B in a dose-dependent manner (Supplementary Fig. 3b, c).

Endogenous INPP4B was diffusely distributed in the cytosol of MCF-7 cells (Supplementary Fig. 3d). However, removal of cytoplasmic proteins by saponin treatment revealed a pool of INPP4B that in part co-localized with CD63-positive late endosomes, the site of Rab7 intracellular localization, and this was most apparent following EGF stimulation (Supplementary Fig. 3e). INPP4B localization to late endosomes was further analyzed by immuno-electron microscopy, which revealed GFP-INPP4B localized to the limiting and internal membranes of late endosomes (Fig. 3d), a site where previous studies have identified both PI(3,4)P$_2$ and PI(3)P[28,29]. In saponin-treated MCF-7 cells, endogenous INPP4B was observed to co-localize with HA-Rab7$^{WT}$ or HA-Rab7$^{Q67L}$ on CD63-positive late endosomes (Fig. 3e). Expression of HA-Rab7$^{T22N}$, a dominant-negative mutant that does not localize to late endosomes (Supplementary Fig. 3f), also prevented the recruitment of endogenous INPP4B to this compartment (Fig. 3e), suggesting that active GTP-bound Rab7 is required for INPP4B subcellular localization to late endosomes.

As Rab7 enhances INPP4B 4-phosphatase activity towards PI(3,4)P$_2$, we predicted that INPP4B may promote PI(3,4)P$_2$ conversion to PI(3)P on late endosomes. PI(3)P resides on early and late endosome membranes, and was detected here using a 2xFYVE/Hrs recombinant biosensor[28]. GFP-INPP4B significantly increased intracellular PI(3)P levels (1.8-fold) (Fig. 3f). PI(3,4)P$_2$, which also resides on early and late endosomes[18,29], was examined by antibody staining in MCF-7 cells but specific staining was not apparent, we assume due to either low signal or antibody affinity. However, MCF-7 cells that expressed *INPP4B* shRNAs (Supplementary Fig. 3g, h) showed more prominent intracellular PI(3,4)P$_2$ puncta suggesting that *INPP4B* depletion suppressed PI(3,4)P$_2$ degradation on endosomes resulting in its accumulation (Fig. 3g). To examine the site of PI(3,4)P$_2$ conversion to PI(3)P by INPP4B, 2xFYVE/Hrs co-localization with the early endosome marker EEA1 and the late endosome

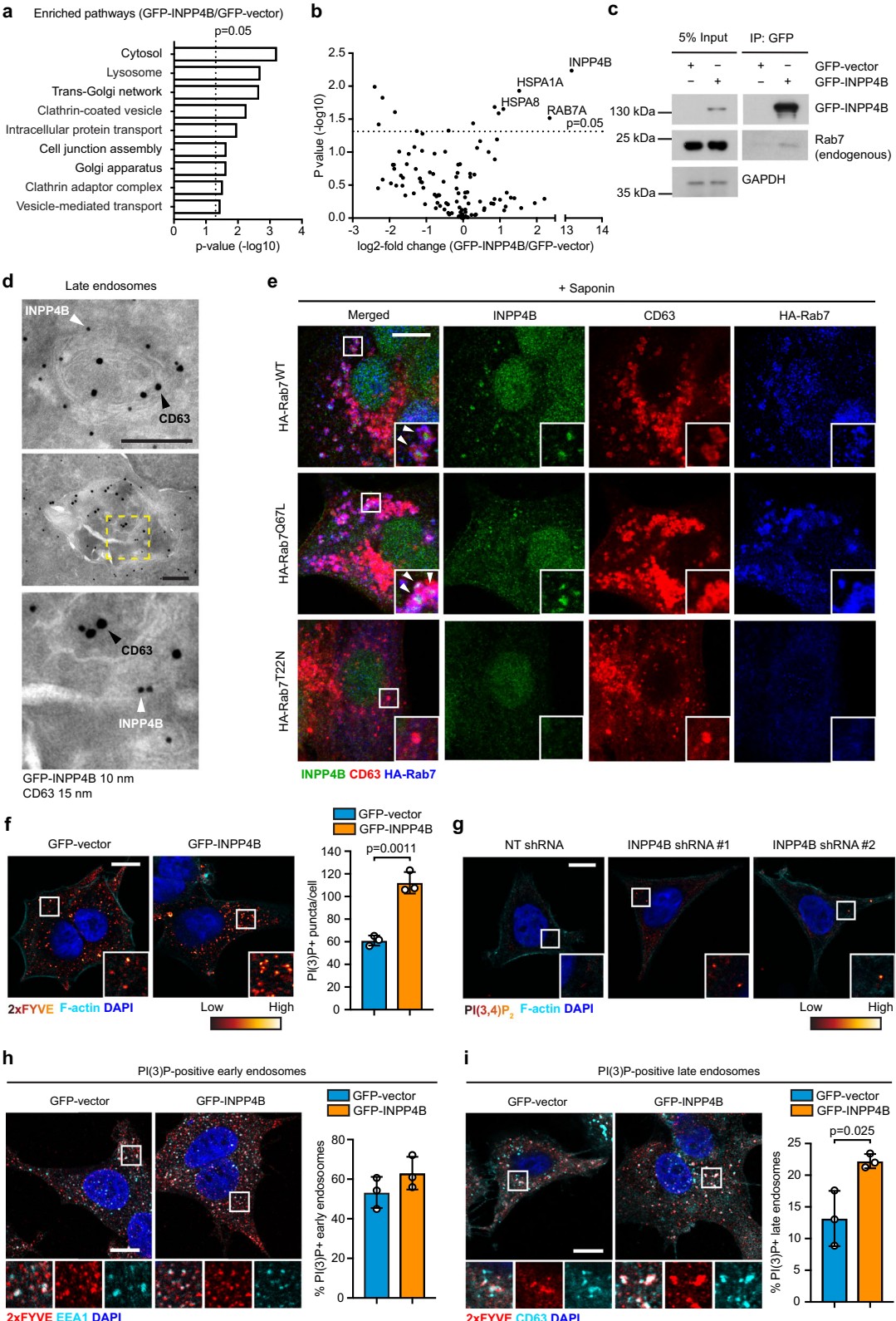

marker CD63 was investigated, revealing GFP-INPP4B did not alter the proportion of PI(3)P-positive early endosomes, but significantly increased PI(3)P-positive late endosomes (1.7-fold) (Fig. 3h, i). In contrast, *INPP4B* shRNA depletion decreased PI(3)P-positive late endosomes (Supplementary Fig. 3i), supporting a role for INPP4B in promoting the conversion of $PI(3,4)P_2$ to PI(3)P on late endosomes.

**INPP4B promotes PI3Kα-dependent late endosome formation.** Endomembrane PI(3)P signaling is essential for binding and localizing FYVE or phox homology (PX) domain-containing proteins to endosomal membranes to regulate numerous processes including early endosome fusion, early-to-late endosome maturation, and cargo sorting[30]. Examination of the endolysosomal compartments of MCF-7 cells showed that GFP-INPP4B

**Fig. 3 INPP4B promotes localized PI(3,4)P$_2$ conversion to PI(3)P on late endosomes. a** MCF-7 cells expressing GFP-INPP4B or GFP-vector were lysed, then subjected to whole cell proteomic analysis by LC-MS/MS. DAVID functional annotation bioinformatics microarray analysis was performed on upregulated proteins in GFP-INPP4B expressing cells versus GFP-vector cells ($n = 3$ experiments). **b** MCF-7 cells expressing GFP-INPP4B or GFP-vector were serum-starved overnight then stimulated with EGF (100 ng/mL) for 5 min. Cells were lysed and subjected to immunoprecipitation using GFP-Trap beads. Bound proteins were identified by mass spectrometry using a cut-off of >2-fold change and $p < 0.05$ ($n = 2$ experiments). **c** MCF-7 cells expressing GFP-INPP4B or GFP-vector were lysed and subjected to immunoprecipitation using GFP-Trap beads. Bound fractions and soluble lysates (5% of input) were subjected to immunoblotting to detect endogenous Rab7. **d** MCF-7 cells expressing GFP-INPP4B were serum starved overnight then stimulated with EGF (100 ng/mL) for 15 min. Cells were fixed and subjected to immuno-electron microscopy analysis using GFP (10 nm) and CD63 (15 nm) antibodies. Representative electron micrographs are shown. Arrows show GFP-INPP4B (white) and CD63 (black) localization. Yellow box indicates area where higher magnification micrograph was captured. **e** MCF-7 cells expressing HA-Rab7$^{WT}$, HA-Rab7$^{Q67L}$, or HA-Rab7$^{T22N}$ were serum-starved overnight then stimulated with EGF (100 ng/mL) for 15 min. Cells were treated with saponin (0.02% w/v) to remove cytoplasmic proteins, then fixed and immunostained using INPP4B, CD63, and HA antibodies. **f** MCF-7 cells expressing GFP-INPP4B or GFP-vector were fixed and immunostained using purified recombinant GST-2xFYVE, DAPI, and phalloidin. Data represent the number of 2xFYVE+ puncta per cell ± SD ($n = 3$ experiments, >50 cells per experiment). **g** MCF-7 cells expressing NT, *INPP4B* #1, or *INPP4B* #2 shRNA were fixed and immunostained with PI(3,4)P$_2$ antibodies, DAPI, and phalloidin. **h, i** MCF-7 cells expressing GFP-INPP4B or GFP-vector were fixed and immunostained using recombinant GST-2xFYVE and EEA1 (**h**) or CD63 (**i**) antibodies, and co-stained with DAPI. Data represent the percentage of 2xFYVE+ early endosomes (**h**) or late endosomes (**i**) ±SD ($n = 3$ experiments, >30 cells per experiment). The inset panels at the lower right or under each image are higher power regions of the boxed areas. Scale bar is 200 nm (**d**), 10 μm (**e–i**). $p$ values determined by a modified Fisher's exact test are indicated in **a**, or by two-tailed unpaired $t$-test in **b**, **f**, **i**.

did not affect EEA1-positive early endosomes, but significantly increased the number of CD63-positive late endosomes (2.5-fold) and LAMP1-positive lysosomes (1.7-fold) (Fig. 4a, b), suggesting that INPP4B promotes the formation of late endosomes/lysosomes. Ultrastructural analysis using electron microscopy was undertaken using BSA-gold labeling of endolysosomal compartments of MCF-7 cells. GFP-INPP4B increased the number of BSA-positive vacuolar structures that resembled late endosomes (Fig. 4c). In contrast, *INPP4B* shRNA depletion reduced late endosomes (Supplementary Fig. 4a–c), suggesting INPP4B regulates late endosome formation. Cargo trafficking to lysosomes was examined using dye-quenched BSA (BSA-DQ) that is incorporated into endosomes by fluid-phase endocytosis, where lysosomal hydrolases subsequently promote de-quenching and activation of the fluorescent signal. GFP-INPP4B significantly increased the number of BSA-DQ-positive structures (2-fold) (Fig. 4d, e), revealing INPP4B enhances the trafficking of endocytosed cargo via late endosomes to the lysosome.

PI(3)P-dependent late endosome formation is catalyzed by Vps34[31]. This class III PI3K phosphorylates PI to PI(3)P. However, our findings suggest PI(3,4)P$_2$ conversion to PI(3)P by INPP4B also contributes to late endosome formation in breast cancer cells. In agreement with this, expression of Myc-INPP4B$^{WT}$ but not PI(3,4)P$_2$ phosphatase-dead Myc-INPP4B$^{C842A}$ increased the number of CD63-positive late endosomes (1.5-fold) in MCF-10A cells (Supplementary Fig. 4d, e). Endosomal PI(3,4)P$_2$ is generated by the sequential actions of class I PI3K, which synthesizes PI(3,4,5)P$_3$, and 5-phosphatases that hydrolyze PI(3,4,5)P$_3$ to PI(3,4)P$_2$, which is then trafficked to early endosomes[18,32]. Endosomal PI(3,4)P$_2$ is also generated from PI(4)P by the class II PI3K PI3KC2α, or the liver-specific PI3KC2γ[33,34]. To determine the upstream pathways that generate the pool of PI(3,4)P$_2$ degraded by INPP4B, we used shRNA or inhibitors of candidate PI3Ks. Depletion of *PIK3C2A* using two distinct shRNAs (Supplementary Fig. 5a, b) did not affect late endosome formation in GFP-vector or GFP-INPP4B MCF-7 cells (Supplementary Fig. 5c, d). In contrast, inhibition of class I PI3K signaling using the pan-class I PI3K inhibitor BKM120 (Supplementary Fig. 5e), the PI3Kα-specific inhibitor BYL719 (Supplementary Fig. 5f) or *PIK3CA* depletion using two distinct siRNAs (Supplementary Fig. 5g, h) suppressed the increased late endosome formation observed in GFP-INPP4B cells (Fig. 4f, g and Supplementary Fig. 5i, j). These results suggest a model whereby class I PI3K signaling facilitates late endosome

formation by generating PI(3,4)P$_2$, which is then hydrolyzed by INPP4B to PI(3)P on late endosomes.

**INPP4B promotes late endosome-mediated cell proliferation and tumor growth.** PI(3)P signaling regulates late endosome formation by recruiting Hrs, a member of the endosomal sorting complex required for transport (ESCRT), that facilitates intraluminal vesicle (ILV) formation[35]. *Hrs* shRNA depletion attenuates the proliferation and xenograft tumor growth of HeLa cells[36]. We hypothesized that by generating PI(3)P on late endosomes, INPP4B promotes late endosome formation via Hrs. Notably shRNA-mediated depletion of Hrs (Supplementary Fig. 6a, b), resulted in fewer CD63-positive late endosomes in GFP-vector cells and reverted the increased late endosome formation in GFP-INPP4B cells to GFP-vector control levels (Fig. 5a, b). This was confirmed at the ultrastructural level by transmission electron microscopy (Supplementary Fig. 6c), revealing that INPP4B contributes to PI(3)P-driven Hrs-mediated late endosome formation.

We also investigated whether INPP4B was dependent on Hrs-mediated late endosome formation for the observed increased proliferation of MCF-7 cells. In anchorage-independent cell growth assays, *Hrs* shRNA depletion had no effect on the size of GFP-vector soft agar colonies, but reduced the increased colony size of GFP-INPP4B cells consistent with Hrs-dependent cell proliferation (Fig. 5c, d). To test Hrs-dependence in vivo for xenograft tumor growth, MCF-7 cells were injected into the fourth inguinal mammary fat pad of female *BALB/c* nude mice. GFP-INPP4B expression significantly increased in vivo tumor volume and ex vivo tumor weight (3-fold) compared to GFP-vector (Fig. 5e–g). Although *Hrs* knockdown did not affect the size or weight of GFP-vector tumors, the enhanced growth of GFP-INPP4B tumors was suppressed by *Hrs* depletion (Fig. 5e–g), indicating that INPP4B promotes the proliferation and tumor growth of *PIK3CA*-mutant ER$^+$ breast cancer cells in an Hrs-dependent manner.

**INPP4B promotes PI3K-dependent Wnt/β-catenin signaling by enhancing lysosomal degradation of GSK3β.** Late endosome trafficking has the potential to affect a number of different pro-proliferative pathways by facilitating activation or degradation of endocytosed signaling receptors and effector proteins. To address this, we examined known cancer pathways in GFP-INPP4B

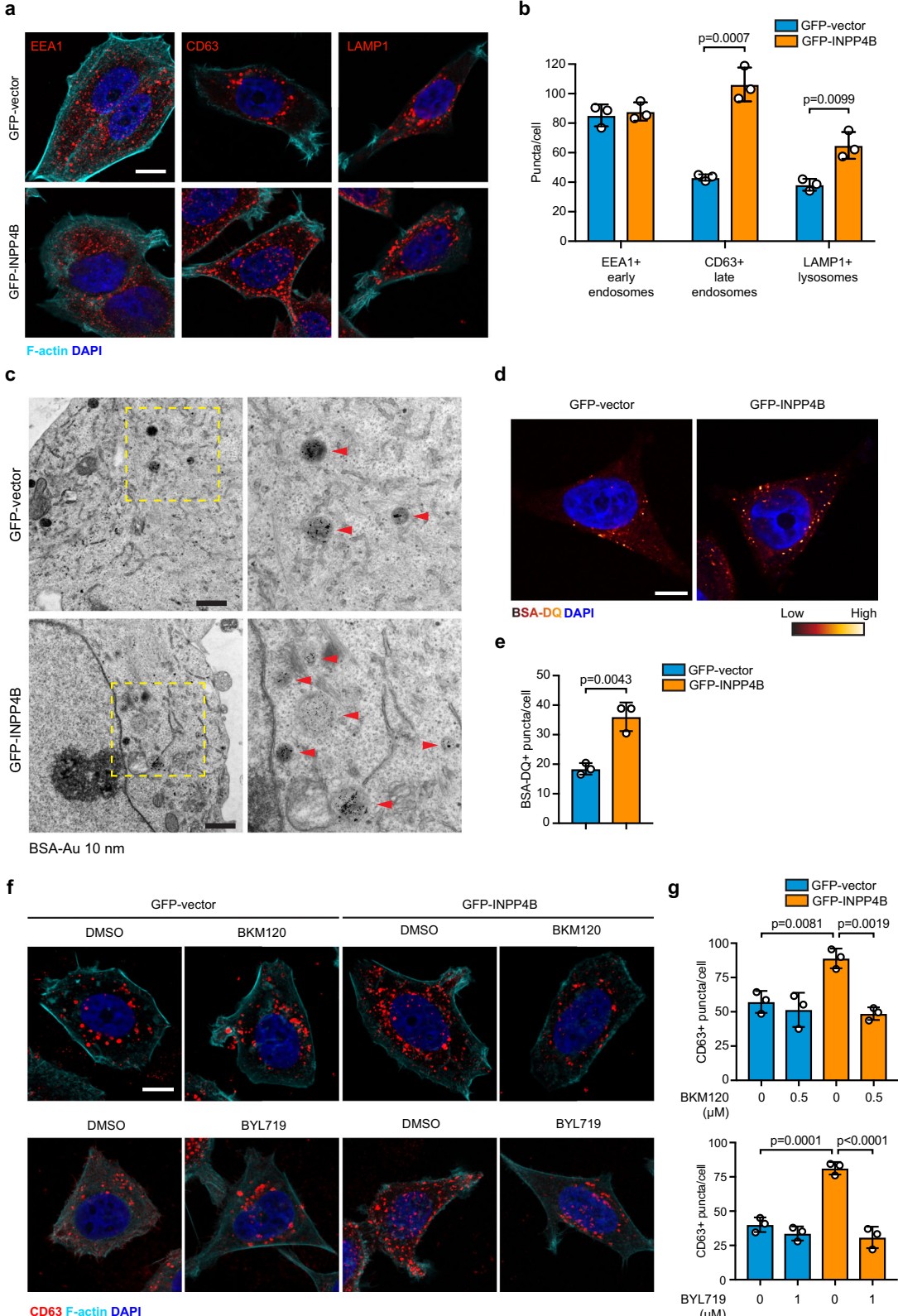

versus GFP-vector expressing MCF-7 cells by nanoString RNA profile analysis, which revealed altered expression of several Wnt/β-catenin pathway genes (Fig. 6a). Wnt/β-catenin signaling regulates mammary epithelial cell proliferation and migration and is hyperactivated in >50% of human breast cancers[37,38]. Increased mRNA expression of the Wnt/β-catenin target genes *AXIN2* (1.8-fold), *LEF1* (5.6-fold), *DKK1* (3.3-fold), and *MYCN* (2-fold) in

GFP-INPP4B-MCF-7 cells was confirmed by quantitative RT-PCR, indicative of increased Wnt/β-catenin signaling (Fig. 6b). Increased mRNA expression of *AXIN2* (1.9-fold), *LEF1* (2.3-fold), and *DKK1* (3.2-fold) was also observed in T47D cells expressing GFP-INPP4B (Supplementary Fig. 7a), demonstrating that INPP4B enhances Wnt/β-catenin signaling in *PIK3CA*-mutant ER+ breast cancer cells. *AXIN2* mRNA levels were increased in

**Fig. 4 INPP4B promotes PI3Kα-dependent late endosome formation. a**, **b** MCF-7 cells expressing GFP-INPP4B or GFP-vector were fixed and immunostained using EEA1, CD63, or LAMP1 antibodies, and co-stained with DAPI and phalloidin (**a**). Data represent the number of EEA1 +, CD63 +, or LAMP1 + puncta per cell ± SD ($n = 3$ experiments, >50 cells per experiment) (**b**). **c** MCF-7 cells expressing GFP-INPP4B or GFP-vector were serum-starved for 1 h, then growth media with BSA-gold (10 nm) was added for 3 h. Cells were fixed and subjected to electron microscopy. Representative electron micrographs of lower and higher magnification are shown. Yellow boxes indicate area where higher magnification micrographs were captured. **d**, **e** MCF-7 cells expressing GFP-INPP4B or GFP-vector were incubated with BSA-DQ (40 µg/mL) for 5 h, then fixed and stained with DAPI (**d**). Data represent the number of BSA-DQ + puncta per cell ± SD ($n = 3$ experiments, >50 cells per experiment) (**e**). **f**, **g** MCF-7 cells expressing GFP-INPP4B or GFP-vector were treated with BKM120 (0.5 µM) or BYL179 (1 µM) for 24 h, or DMSO as a vehicle control, then fixed and stained with CD63 antibodies and co-stained with DAPI and phalloidin (**f**). Data represent the number of CD63 + puncta per cell ± SD ($n = 3$ experiments, >50 cells per experiment) (**g**). Scale bar is 10 µm (**a**, **d**, **f**), 2 µm (**c**). $p$ values determined by two-tailed unpaired $t$-test are indicated in **b**, **e**, or by one-way ANOVA with Tukey post hoc test in **g**.

Myc-INPP4B$^{WT}$ but not Myc-INPP4B$^{C842A}$ expressing MCF-10A cells (Supplementary Fig. 7b), suggesting that increased Wnt/β-catenin activation is dependent on INPP4B PI(3,4)P$_2$ 4-phosphatase activity.

Wnt/β-catenin pathway activation requires inhibition of GSK3β, a member of the β-catenin destruction complex (AXIN, APC, CK1, PP2A, and β-TrCP). In the absence of Wnt/β-catenin activation, GSK3β phosphorylates β-catenin$^{S33/S37/T41}$ leading to β-catenin proteasomal degradation which results in inhibition of Wnt/β-catenin signaling. When Wnt/β-catenin signaling is activated, GSK3β is recruited to endosomes where it is sequestered within the lumen of late endosomes and degraded by lysosomes[39,40]. This allows β-catenin to accumulate in the nucleus where it binds TCF/LEF transcription factors and promotes Wnt target gene transcription. Wnt/β-catenin signaling was activated by Wnt3a and R-spondin treatment, and under these conditions GFP-INPP4B cells exhibited increased AXIN2 mRNA expression and non-phospho-β-catenin$^{S33/S37/T41}$ (active-β-catenin) levels compared to GFP-vector controls (Fig. 6c, d). In addition, GFP-INPP4B expression reduced GSK3β protein levels under basal and Wnt-stimulated conditions, consistent with enhanced destruction of GSK3β (Fig. 6d). No changes in GSK3B mRNA levels were observed in GFP-INPP4B cells (Supplementary Fig. 7c). Decreased GSK3β protein levels were restored upon inhibition of late endosome formation by Hrs shRNA depletion (Fig. 6f, g). GSK3β sequestration within late endosomes was examined using a protease protection assay, where following digitonin permeabilization, cytoplasmic proteins are sensitive to proteinase K degradation but proteins within organelles are protected[41]. GSK3β is not readily detectable in late endosomes in the absence of Wnt stimulation[40,42], however, GFP-INPP4B but not GFP-vector cells exhibited clear GSK3β protease protection consistent with enhanced GSK3β trafficking to late endosomes (Fig. 6h). This was confirmed by immunofluorescence where co-localization of RFP-GSK3 with LysoTracker, a dye that accumulates in late endosomes/lysosomes, was observed in GFP-INPP4B but not GFP-vector cells (Fig. 6i). Together, this data suggests that INPP4B promotes GSK3β trafficking and degradation via the late endosome-lysosome compartments.

To assess whether INPP4B promotes the proliferation of PIK3CA-mutant ER$^+$ breast cancer cells via Wnt/β-catenin signaling, assays were performed in the presence of the small molecule Porcupine (Porcn) inhibitors IWP-2 or LGK-974, which inhibit Wnt ligand secretion, or the Tankyrase (TNKS) inhibitor IWR-1-endo, which promotes constitutive β-catenin destruction complex stabilization[43]. Inhibitors were titrated to determine the maximum concentration that had no effect on the proliferation of MCF-7 cells expressing GFP-vector. Treatment of GFP-INPP4B MCF-7 cells with IWP-2, LGK-974, or IWR-1 reduced cell proliferation to a level comparable to vehicle-treated GFP-vector cells (Supplementary Fig. 7d, e). Furthermore, in anchorage-independent cell growth assays, IWP-2 treatment reduced the colony size of GFP-INPP4B cells (Fig. 6j, k).

Previously reported gene expression analysis of 249 ERα$^+$ breast cancers revealed upregulation of several Wnt pathway genes (TCF7L2, MSX2, WNT5A, and TNFRSF11B) in PIK3CA-mutant versus wild-type tumors[44], suggesting there may be a functional link between PI3Kα and Wnt signaling in ER$^+$ breast cancer. Using the METABRIC dataset, we assessed the expression of a panel of 17 common Wnt/β-catenin pathway genes, including several Wnt target genes, in 1459 ER$^+$ breast cancers (Fig. 7a). PIK3CA-mutant cancers displayed higher expression of 12 of the 17 Wnt genes compared to PIK3CA wild-type tumors, including five Wnt target genes (LEF1, MYCN, FZD7, SFRP2, and TNFRSF11B) and seven Wnt signaling components (TCF7L1, TCF7L2, CTNNB1, FZD4, SFRP1, WNT5A, and MSX2). We also examined the expression of these genes in a separate cohort of 698 luminal (ER/PR$^+$) breast cancers using the TCGA dataset (Supplementary Fig. 8), where PIK3CA-mutant cancers displayed significantly higher expression of eight Wnt genes (LEF1, TCF7L1, TCF7L2, CTNNB1, LRP6, SFRP2, WNT5A, and MSX2). Therefore PIK3CA-mutant ER$^+$ breast cancers, which exhibit increased INPP4B expression, display a gene expression profile consistent with increased Wnt/β-catenin signaling. To determine whether INPP4B promotes crosstalk between PI3K and Wnt/β-catenin pathways, we treated GFP-vector and GFP-INPP4B MCF-7 cells with increasing doses of the PI3K inhibitor BKM120 (0.5 and 1 µM) and measured the mRNA expression of the Wnt target genes AXIN2, LEF1, and MYCN (Fig. 7b–d). In GFP-vector cells, a low dose of BKM120 had minimal effect on Wnt target gene expression, whereas at a higher dose Wnt gene expression was reduced. In GFP-INPP4B expressing cells, a low BKM120 partially rescued the increased Wnt gene expression, and this was further reduced at a higher concentration. Together, these findings are consistent with an interpretation that INPP4B promotes Wnt/β-catenin signaling and thereby the proliferation of PIK3CA-mutant ER$^+$ breast cancer cells via enhanced late endosome formation (Fig. 7e).

## Discussion

Here, we identify that INPP4B promotes late endosome/lysosome formation and trafficking that activates Wnt/β-catenin signaling in PIK3CA-mutant ER$^+$ breast cancer. Increased INPP4B mRNA and protein expression was detected in 14–40% of breast cancers associated with ER-positivity and mutant PIK3CA expression. INPP4B increased the proliferation and tumor growth of PIK3CA-mutant ER$^+$ breast cancer cells, despite reduced AKT activation. INPP4B promoted PI(3,4)P$_2$ conversion to PI(3)P on late endosomes, acting downstream of PI3Kα to increase late endosome formation and trafficking. In this context, INPP4B increased the lysosomal degradation of GSK3β leading to enhanced Wnt/β-catenin signal activation. We therefore propose INPP4B facilitates crosstalk between PI3K and Wnt/β-catenin signaling pathways on a late endosome signaling hub in PIK3CA-mutant ER$^+$ breast cancer.

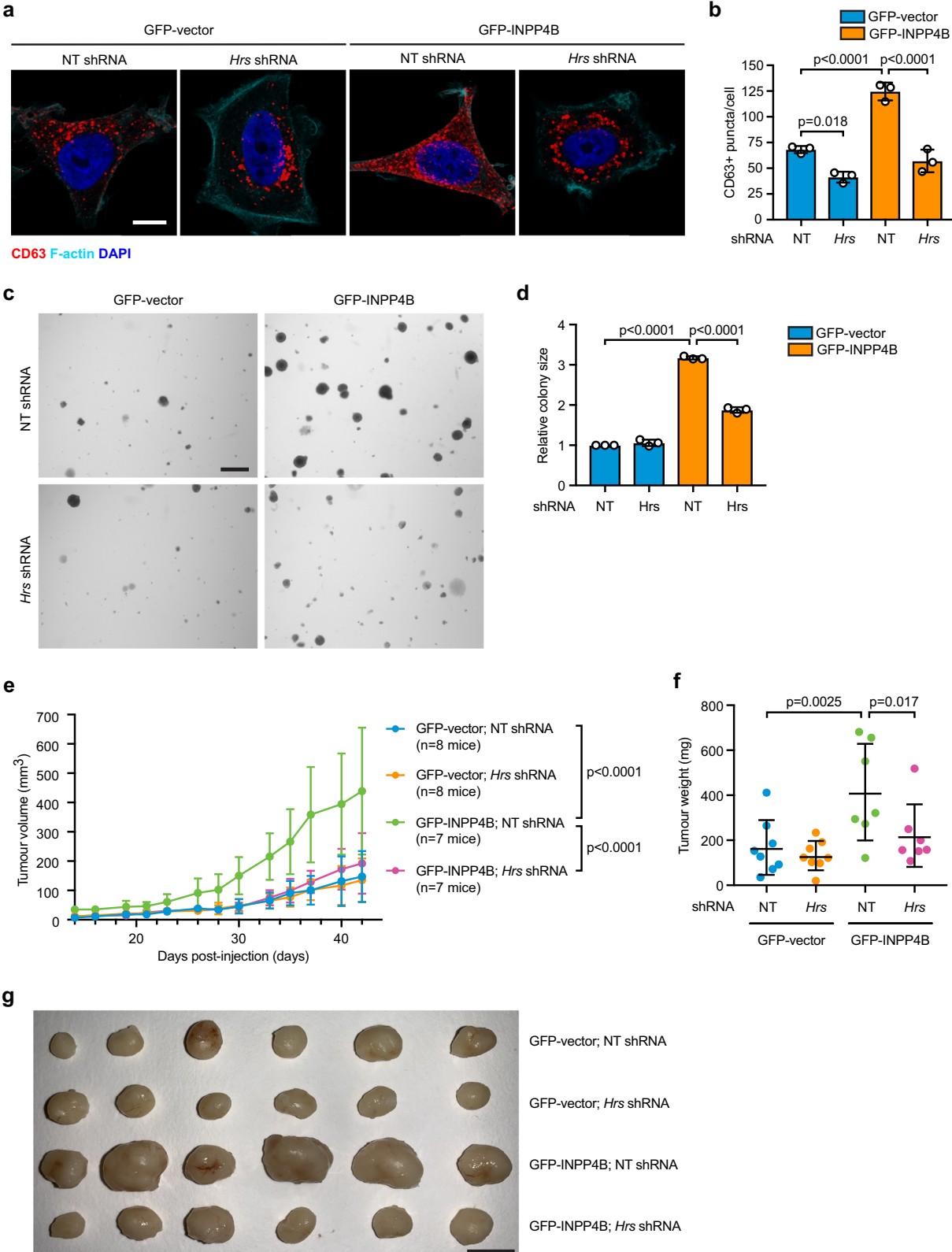

PIK3CA mutations are common in ER[+] tumors[7,8], however, these mutations are weak oncogenes in murine models of ER[+] breast cancer, with long tumor latency and variable penetrance in vivo[45,46]. Interestingly PIK3CA-mutant breast cancers also display low dependence on AKT signaling for unknown reasons[4,12]. Our discovery that INPP4B promotes oncogenic signaling in PIK3CA-mutant ER[+] breast cancer, despite suppressing AKT signaling, suggests Wnt/β-catenin rather than AKT activation is responsible for the hyper-proliferation. INPP4B loss, which occurs most frequently in triple-negative breast cancer where PIK3CA mutations are rare, enhances oncogenic PI3K/AKT signaling[5,6]. This suggests that INPP4B plays a dual role as an oncogene and tumor suppressor in breast cancer via activation of alternate PI3K-dependent signaling pathways. We propose this

**Fig. 5 INPP4B promotes late endosome-dependent cell proliferation and tumor growth. a**, **b** MCF-7 cells co-expressing GFP-INPP4B or GFP-vector, and NT or *Hrs* shRNA, were fixed and immunostained using CD63 antibodies, and co-stained with DAPI and phalloidin (**a**). Data represent the number of CD63 + puncta per cell ± SD (*n* = 3 experiments, >50 cells per experiment) (**b**). **c**, **d** MCF-7 cells co-expressing GFP-INPP4B or GFP-vector, and NT or *Hrs* shRNA, were suspended in 0.3% soft agar and cultured for 4 weeks to allow anchorage-independent colony growth (**c**). Data represent the relative colony size ± SD (*n* = 3 experiments in triplicate, *n* > 50 colonies/experiment) (**d**). **e**–**g** 1 × 10^6 MCF-7 cells co-expressing GFP-vector;NT shRNA (*n* = 8 mice), GFP-vector:*Hrs* shRNA (*n* = 8 mice), GFP-INPP4B;NT shRNA (*n* = 7 mice), or GFP-INPP4B;*Hrs* shRNA (*n* = 7 mice) were mixed with Matrigel and injected into the fourth mammary fat pad of female BALB/c nude mice. From 2 weeks postinjection, tumor size was measured using callipers three times a week. Data represent tumor size measured from 2 weeks postinjection over the course of the experiment ±SD (**e**). Data represent extracted mammary tumor weight ex vivo at 42 days postinjection ±SD (**f**). Comparison of six representative extracted mammary tumors ex vivo at 42 days postinjection (**g**). Scale bar 10 μm (**a**), 1 mm (**c**), 1 cm (**g**). *p* values determined by one-way ANOVA with Tukey post hoc test are indicated in **b**, **d**, by two-way ANOVA with Tukey post hoc test in **e**, or by one-way ANOVA in **f**.

is due to INPP4B-mediated degradation of one phosphoinositide signal, PI(3,4)P$_2$, which activates oncogenic Akt signaling, to generate another signal, PI(3)P, which we show here activates Wnt signaling. Interestingly, INPP4B also enhanced Wnt/β-catenin activation and proliferation of MCF-10A cells, which are ER$^-$ and express wild-type PI3Kα, suggesting that ER-positivity or mutant-*PIK3CA* are dispensable for INPP4B oncogenic signaling although these factors have the potential to enhance signaling. In ER$^+$ breast cancer, reduced dependence on AKT signaling by *PIK3CA*-mutant breast cancers has been associated with increased SGK3 signaling[4], which can also be activated by INPP4B[23]. INPP4B can also regulate AKT activation via PTEN destabilization in colon cancer[21,23]. However, we found no evidence of enhanced SGK3 activation or PTEN degradation in INPP4B-overexpressing MCF-7 cells under the conditions used in our experiments, which may relate to the relative levels of INPP4B overexpression or specific agonist stimulus used to activate these signaling pathways.

Here we demonstrated endogenous and ectopically-expressed INPP4B localized to late endosomes of ER$^+$ breast cancer cells. INPP4B interacted with Rab7 which facilitated INPP4B recruitment to late endosomes and increased its PI(3,4)P$_2$ 4-phosphatase activity. These findings are reminiscent of the related 4-phosphatase, INPP4A, that binds the early endosome protein Rab5, which activates INPP4A PI(3,4)P$_2$ 4-phosphatase activity to regulate the early stages of endocytosis[32,47]. In ER$^+$ breast cancer cells, INPP4B functions at the later stages of endosomal trafficking to regulate Hrs-dependent late endosome formation, suggesting that distinct PI(3,4)P$_2$ to PI(3)P conversion events regulate different stages of endosomal trafficking. As INPP4A is not expressed in breast cancer cells[6], endosomal PI(3,4)P$_2$ to PI(3)P conversion in breast cancer is most likely dependent on INPP4B. A recent report found INPP4B inhibits receptor tyrosine kinase (RTK) trafficking from early to late endosomes in MCF-10A mammary cells by unknown mechanisms[13], and it is possible the role INPP4B plays in late endosome formation and trafficking contributes to RTK trafficking. Other reports have shown INPP4B is enriched on early endosomes of thyroid cancer cells, where it suppressed localized AKT2 activation[16,18]. We cannot exclude the possibility that INPP4B localization to early versus late endosomes may be cell type dependent or defined by yet to be determined interactions.

Although endosomal PI(3)P synthesis is primarily attributed to phosphorylation of PI by the Class III PI3K, Vps34[31], our findings demonstrate that PI(3,4)P$_2$ hydrolysis by INPP4B is an alternate mechanism of PI(3)P synthesis on late endosomes. A number of other studies have shown PI(3,4)P$_2$ and PI(3)P levels on endosomes are partially dependent on class I and class II PI3K signaling[18,29,32,33]. PI(3,4)P$_2$ is locally synthesized by PI3KC2α on endocytic pits during clathrin-mediated endocytosis[33], but endosomal PI(3,4)P$_2$ synthesis also requires activation of class I

PI3K signaling at the plasma membrane[18,29,32]. Here we identified INPP4B-mediated late endosome formation was PI3Kα-dependent, but independent of PI3KC2α, suggesting PI3Kα is the primary source of PI(3,4)P$_2$ on late endosomes, that is in turn hydrolyzed by INPP4B at this site. As PI(3,4,5)P$_3$ synthesis occurs primarily at the plasma membrane, class I PI3K-dependent PI(3,4)P$_2$ may be generated locally by 5-phosphatases and the lipid is internalized via endocytosis, or generated by 5-phosphatases on endomembranes. Spatiotemporal analysis of class I PI3K-generated PI(3,4)P$_2$ signals is required to elucidate this molecular pathway in future studies.

Hyperactivation of Wnt/β-catenin signaling occurs in 50% of breast cancers[37], but this rarely occurs via mutations in Wnt pathway components, although this remains the dominant mechanism of activation in other cancers[38]. Late endosomes and lysosomes are critical components of Wnt/β-catenin signaling as this endosomal compartment sequesters and degrades GSK3β, preventing its phosphorylation and inactivation of β-catenin[40]. Endosome acidification, MITF-dependent lysosome biogenesis, and ESCRT proteins Hrs and Vps4 are all required for Wnt/β-catenin activation[40,48–51], but the role of late endosome-dependent Wnt activation in cancer pathogenesis has not been well defined. Notably, depletion of *Hrs*, a known regulator of Wnt/β-catenin signaling[40], rescued INPP4B-driven in vivo tumor growth consistent with an interpretation that pathway activation occurs at the level of late endosome formation. Although we cannot exclude the possibility that INPP4B affects other endosome-dependent signaling pathways, such as EGFR recycling or degradation, the pro-proliferative function of INPP4B was rescued by small molecule Wnt inhibitors demonstrating late endosome-mediated Wnt signaling as an oncogenic mechanism of action of INPP4B. Recent studies have also found that Wnt signaling increases macropinocytosis and the number of endo-lysosomes to promote protein uptake and degradation[42,50]. Macropinocytosis enhances the proliferation and drug resistance of breast cancer cells[52], thus INPP4B regulation of Wnt signaling and lysosome activity has the potential to regulate cancer cell metabolism and should be investigated in future studies. ER$^+$ *PIK3CA*-mutant breast tumors display upregulation of several Wnt pathway genes (*TCF7L2*, *MSX2*, *WNT5A*, and *TNFRSF11B*)[44,53]. Our independent analysis of primary ER$^+$ *PIK3CA*-mutant breast cancers, which exhibit high *INPP4B* expression, identified these and additional upregulated Wnt genes (*LEF1*, *MYCN*, *FZD7*, *SFRP2*, *TCF7L1*, *CTNNB1*, *FZD4*, and *SFRP1*). Crosstalk between class I PI3K and Wnt/β-catenin signaling has been previously reported[54,55], although the mechanisms remain poorly understood. One proposed mechanism of crosstalk is via GSK3β, which is phosphorylated on its N-terminal tail region by AKT, however, GSK3β binding to AXIN in the β-catenin destruction complex prevents AKT-dependent phosphorylation at this site[56]. Our data supports a model whereby

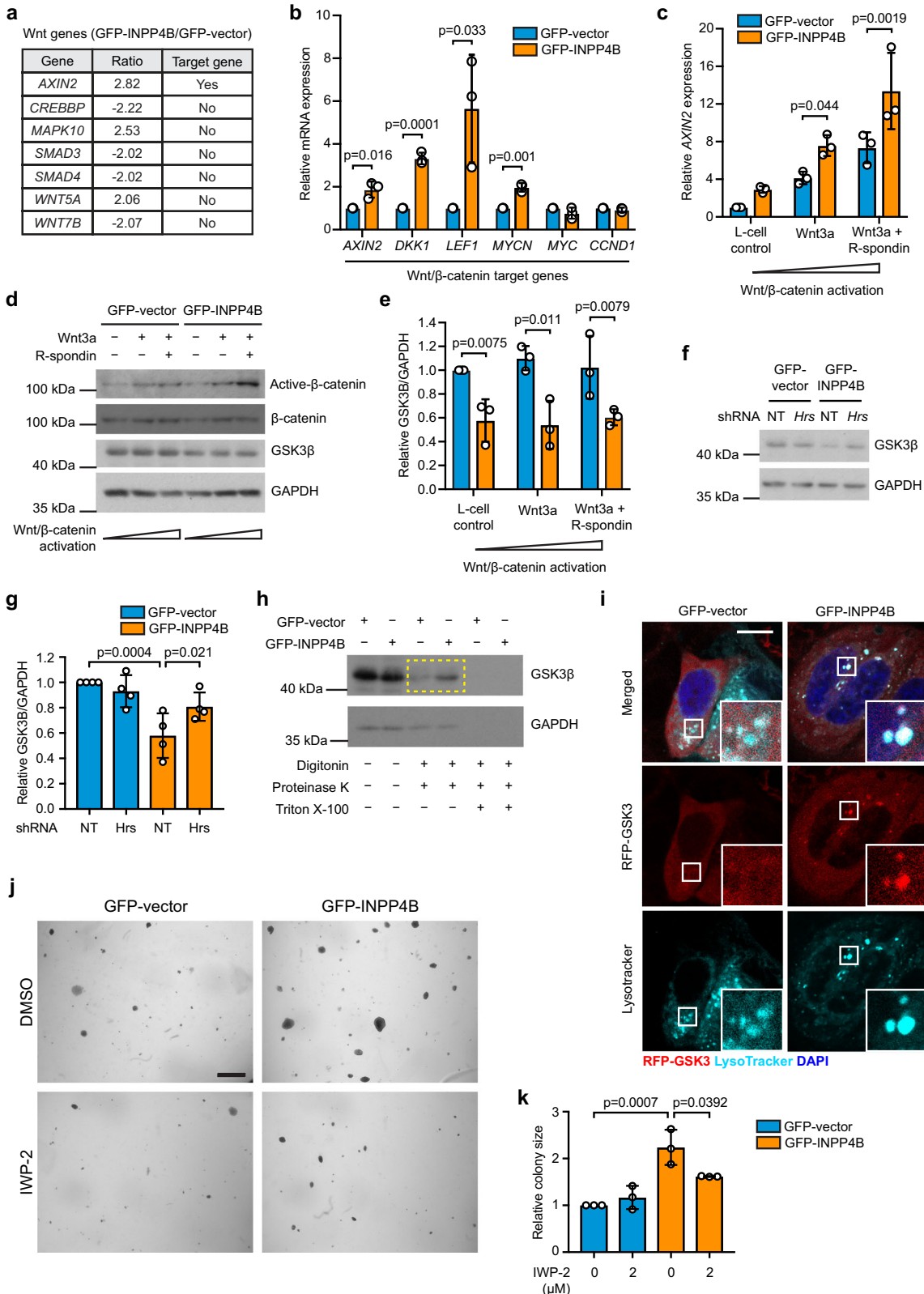

INPP4B regulates crosstalk between PI3K and Wnt/β-catenin pathways by promoting the endosomal sequestration and lysosomal degradation of GSK3β (Fig. 7e).

Endocrine therapy remains the standard first line treatment for hormone receptor-positive breast cancers, although inherent and acquired resistance does occur in a subset of patients leading to advanced disease[57]. *PIK3CA* mutations are the most common

alteration in ER+ breast cancers[7,8], and are a target for emerging PI3K inhibitor therapies. Recently, alpelisib (PI3Kα inhibitor) in combination with fulvestrant (endocrine therapy) was approved for use in patients with *PIK3CA*-mutant, hormone receptor-positive, HER2-negative advanced breast cancer[11]. Double *PIK3CA* mutations predict improved response to taselisib (PI3Kα/γ/δ inhibitor) in combination with fulvestrant in ER+

**Fig. 6 INPP4B promotes Wnt/β-catenin activation. a** Differentially expressed (>2-fold) Wnt pathway genes identified from nanoString RNA profile analysis of GFP-INPP4B versus GFP-vector expressing MCF-7 cells using the nCounter PanCancer Pathways codeset. Wnt/β-catenin target genes are indicated. **b** RNA was extracted from MCF-7 cells expressing GFP-INPP4B or GFP-vector, and two-step quantitative RT-PCR was performed using primers for *AXIN2*, *LEF1*, *DKK1*, *MYCN*, *MYC*, or *CCND1*. Expression was normalized to *GAPDH*. Expression was determined using the ΔΔCt method and expressed relative to GFP-vector control cells (±SD), which were assigned an arbitrary value of 1 (n = 3 experiments). **c–e** MCF-7 cells expressing GFP-INPP4B or GFP-vector were stimulated with 50% Wnt3a-conditioned media ± 10% R-spondin-conditioned media or 50% L-cell control media. **c** RNA was extracted and two-step quantitative RT-PCR was performed using primers for *AXIN2* and expression was normalized to *GAPDH*. Expression was determined using the ΔΔCt method and expressed relative to L-cell control media-treated GFP-vector control cells (±SD) which were assigned an arbitrary value of 1 (n = 3 experiments). **d** Cells were lysed and subjected to immunoblotting with non-phospho β-catenin$^{S33/S37/T41}$ (active-β-catenin), β-catenin or GSK3β antibodies, and GAPDH antibodies were used as loading control. **e** Data represent the mean GSK3β levels relative to GAPDH ± SD (n = 3 experiments). **f, g** MCF-7 cells co-expressing GFP-INPP4B or GFP-vector, and NT or *Hrs* shRNA, were lysed and subjected to immunoblotting with GSK3β antibodies or GAPDH antibodies as a loading control (**f**). Data represent the mean GSK3β levels relative to GAPDH ± SD (n = 4 experiments) (**g**). **h** MCF-7 cells expressing GFP-INPP4B or GFP-vector were permeabilized with 6.5 μg/mL digitonin, then treated with 1 μg/mL proteinase K in the presence or absence of 0.1% Triton X-100. Cells were lysed and immunoblotted with GSK3β antibodies, and GAPDH antibodies were used as loading control. Yellow box indicates GSK3β protein protected from proteinase K treatment. **i** MCF-7 cells co-expressing RFP-GSK3 and GFP-INPP4B or GFP-vector were incubated with 50 nM LysoTracker Deep Red for 1 h, then fixed and stained with DAPI. The inset panels at the lower right of each image are higher power regions of the boxed areas. **j, k** MCF-7 cells expressing GFP-INPP4B or GFP-vector were suspended in 0.3% soft agar in the presence of 2 μM IWP-2, or DMSO as a vehicle control, and cultured for 4 weeks to allow anchorage-independent colony growth (**j**). Data represent the relative colony size ± SD (n = 3 experiments in triplicate, 50 colonies/experiment) (**k**). Scale bar is 10 μm (**i**), 1 mm (**j**). *p* values determined by two-tailed unpaired *t*-test are indicated in **b**, by one-way ANOVA in **c, e, g**, or by one-way ANOVA with Tukey post hoc test in **k**.

metastatic breast cancer[58]. However, the clinical outcomes using PI3K inhibitors have been limited as tumors invariably progress, and develop resistance through a variety of mechanisms including acquired *PTEN* loss or insulin feedback[59,60]. Identification of additional biomarkers and novel combination therapies are therefore essential for improving patient outcomes and slowing disease progression. Our findings that ER$^+$ *PIK3CA*-mutant tumors exhibit increased INPP4B expression leading to enhanced Wnt/β-catenin activation suggest these tumors may benefit from combined treatment with PI3K and Wnt/β-catenin inhibitors.

## Methods
**Tissue Scan Breast Cancer cDNA arrays (I–IV).** Breast cancer cDNA arrays (176 breast cancers and 16 tumor adjacent normal breast tissues) were purchased from OriGene (BCRT101, BCRT102, BCRT103, and BCRT104). qRT-PCR reactions were performed with *INPP4B* and *β-ACTIN* primers (Supplementary Table 1) according to the manufacturer's instructions in an MX3000P qPCR system (Stratagene). *INPP4B* mRNA expression was normalized to *β-ACTIN* using the ΔΔCt method[61]. *INPP4B* expression levels in the breast cancer samples were calculated relative to the mean of the normal tissue samples. Statistical significance was determined by performing a Mann–Whitney test assuming non-Gaussian distribution for two groups or a Kruskal–Wallis test assuming non-Gaussian distribution for more than two groups. Fisher's exact test was used to determine the significance of the contingency between two categorical groups.

**Immunohistochemical analysis of INPP4B expression in primary human breast tissue microarrays.** Primary human breast cancer tissue microarrays (224 breast cancers and 32 tumor-adjacent normal breast tissues) were purchased from US Biomax (BR1921a and BR2082a). Immunohistochemistry was performed using an INPP4B antibody generated by the host laboratory[6]. Paraffin embedded tissues were dewaxed in three changes of xylene then rehydrated in three changes of ethanol. Heat induced antigen retrieval was performed in a pressure cooker for 3 mins at 120 °C in Novocastra Epitope Retrieval solution pH 9 (Leica Microsystems, Cat # RE7119-CE). Sections were incubated in 1% (w/v) BSA, 50 mM Tris pH 8, 150 mM NaCl for 1 h to block nonspecific antibody binding then incubated with INPP4B primary antibodies (Supplementary Table 2) diluted in blocking buffer overnight at 4 °C. Sections were washed three times with 50 mM Tris pH 8, 150 mM NaCl then endogenous peroxidase activity was quenched with 0.3% (v/v) hydrogen peroxide for 10 min. Sections were washed three times in 50 mM Tris pH 8, 150 mM NaCl. Immunoreactivity was detected using the Envision+ HRP mouse kit (Dako, Cat # K4001). Post-primary block was added to slides for 10 min at room temperature. Sections were washed twice in 50 mM Tris pH 8, 150 mM NaCl before HRP conjugated polymer was added for 15 min. Sections were washed twice in 50 mM Tris pH 8, 150 mM NaCl. DAB chromogen was added to the sections for 10 min, and sections were washed in dH$_2$O for 30 s. Slides were counterstained with haematoxylin for 30 s, then rinsed under running tap water for 30 s. Slides were washed with acid alcohol for 2 s, then rinsed under running tap water for 30 s. Slides were then washed 3 times in ethanol for 1 min each, then 3 times in xylene for 1 min each. Cover slips were mounted onto slides using DPX (Labchem, Cat #

AJA3197). Slides were imaged by brightfield microscopy using an Aperio Scanscope microscope (Leica Microsystems), and analyzed using Aperio Imagescope version 12.4.0.5043 software (Leica Microsystems, https://www.leicabiosystems.com) (Monash Histology Platform, Monash University, Australia). INPP4B staining intensity was scored blindly by Prof Catriona McLean. INPP4B staining were scored as negative (0), low (1), medium (2), or high (3) using normal adjacent breast tissue as a control for INPP4B levels. Fisher's exact test was used to determine the significance of the contingency between two categorical groups.

**Cell culture.** MCF-7 (Cat # HTB-22), T47D (Cat # HTB-133), MCF-10A (Cat # CRL-10317), and HEK293T (Cat # CRL-3216) cells were purchased from ATCC. MCF-10A cells were cultured in DMEM/F12 supplemented with 5% (v/v) horse serum, 2 mM L-glutamine, 100 units/mL penicillin, 1% (v/v) streptomycin, 0.5 μg/mL hydrocortisone, 100 ng/mL cholera toxin, 10 μg/mL insulin, and 5 ng/mL EGF. MCF-7 cells were cultured in DMEM supplemented with 10% (v/v) FCS, 2 mM L-glutamine, 100 units/mL penicillin, 1% (v/v) streptomycin, and 10 μg/mL insulin. T47D cells were cultured in RPMI supplemented with 10% (v/v) FCS, 2 mM L-glutamine, 100 units/mL penicillin, 1% (v/v) streptomycin, and 10 μg/mL insulin. HEK293T were cultured in DMEM supplemented with 10% (v/v) FCS, 2 mM L-glutamine, 100 units/mL penicillin and 1% (v/v) streptomycin. All cells were maintained in a 5% CO$_2$ humidified 37 °C incubator. All aseptic culture techniques were performed in a class II biohazard hood. Cells were routinely tested to confirm the absence of mycoplasma contamination. For EGF stimulation, cells were washed once with PBS then incubated overnight with phenol red-free DMEM without FCS. The following day, 100 ng/μL EGF (Gibco, Cat # 354052) diluted in phenol red-free DMEM without FCS was added for the indicated time points.

**Three dimensional MCF-10A acini cell culture.** Three-dimensional culture of MCF-10A cells was carried out in chamber slides as previously described[26]. Eight-well chamber slides were coated with 45 μL of Matrigel (Corning, Cat # 356231) per well which was allowed to set at 37 °C. Cells were detached with trypsin/EDTA solution and suspended in 5 mL of DMEM/F12 supplemented with 20% (v/v) horse serum, 2 mM L-glutamine, 100 units/mL penicillin, and 1% (v/v) streptomycin, then centrifuged at 230 × g for 3 min. After removal of the supernatant, cells were resuspended in DMEM/F12 supplemented with 2% (v/v) horse serum, 2 mM L-glutamine, 100 units/mL penicillin, 1% (v/v) streptomycin, 0.5 μg/mL hydrocortisone, 100 ng/mL cholera toxin, 10 μg/mL insulin, 5 ng/mL EGF, and 2% (v/v) Matrigel with doxycycline (Sigma, Cat # D9891, 1 μg/mL). Cells were seeded into Matrigel coated chamber slides at 5000 cells per well in 400 μL DMEM/F12 supplemented with 2% (v/v) horse serum, 2 mM L-glutamine, 100 units/mL penicillin, 1% (v/v) streptomycin, 0.5 μg/mL hydrocortisone, 100 ng/mL cholera toxin, 10 μg/mL insulin, 5 ng/mL EGF, 1 μg/mL doxycycline, and 2% (v/v) Matrigel. Cells were incubated in a 5% CO$_2$ humidified 37 °C incubator, with media being replaced every 3–4 days.

**Generation of stable cell lines by viral transduction.** To generate pBMN-HA-vector, pBMN-HA-Rab7$^{WT}$, pBMN-HA-Rab7$^{Q67L}$, and pBMN-HA-Rab7$^{T22N}$ constructs, the pBMN-Z retroviral vector (Gary Nolan, Stanford University, Addgene # 1734) was digested with Sal1/BamH1 before cloning using the HiFi DNA Assembly 1232 Kit (New England Biolabs, Cat # E5520S) according to the manufacturer's instructions. pBMN-HA vector, pBMN-HA-Rab7$^{WT}$, pBMN-HA-

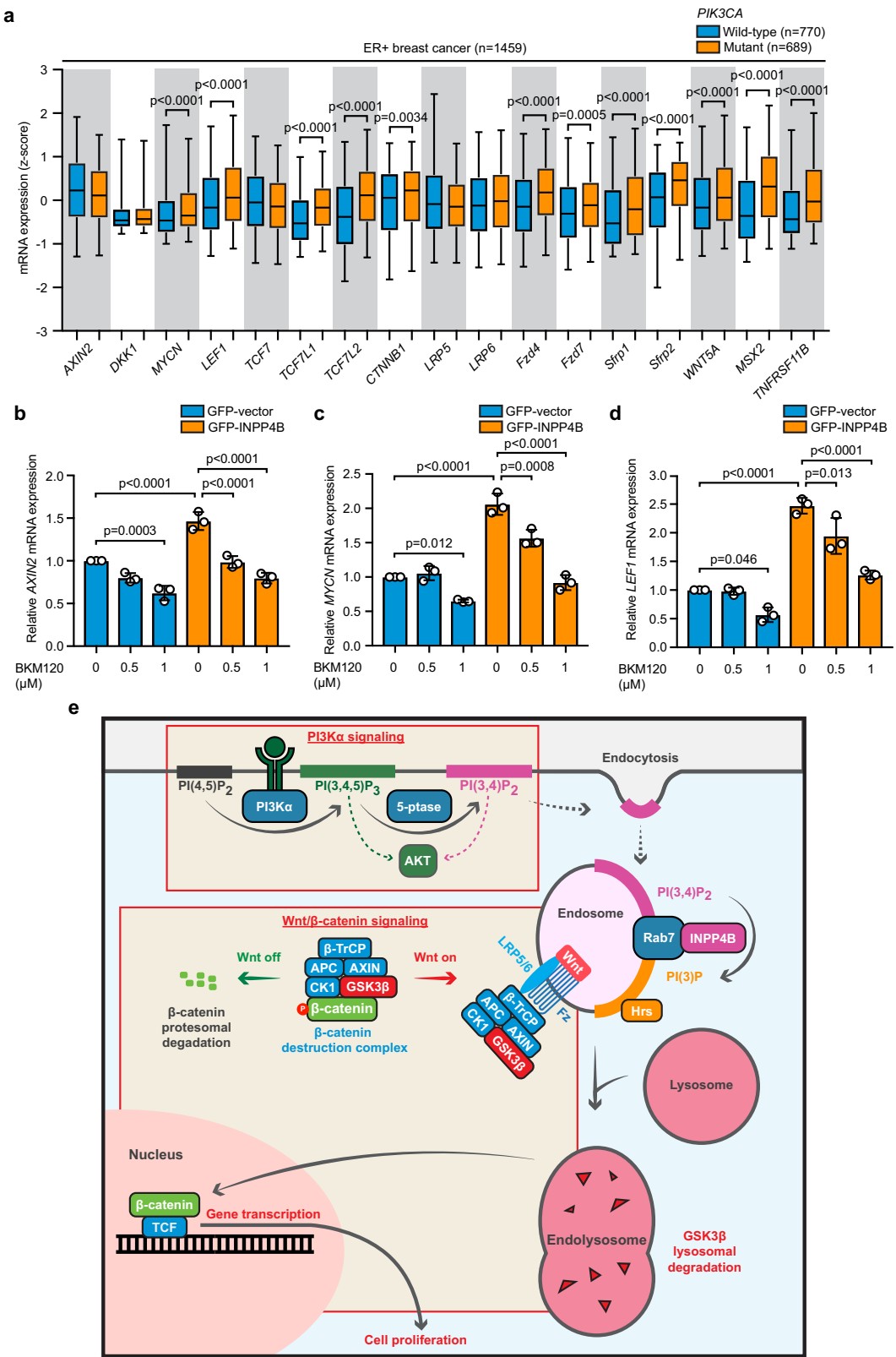

Rab7$^{Q67L}$, and pBMN-HA-Rab7$^{T22N}$ were generated by PCR amplification of pCGN-HA[62], or pMH-SFB-Rab7$^{WT}$, pMH-SFB-Rab7$^{Q67L}$, or pMH-SFB-Rab7$^{T22N}$ (Supplementary Table 1), which were a kind gift from Subbareddy Maddika[63]. pLVX-TRE3G-mCherry-Myc-INPP4B$^{WT}$ and pLVX-TRE3G-mCherry-Myc-INPP4B$^{C842A}$ lentiviral plasmids were generated by digesting pLVX-TRE3G-mCherry (Clontech, Cat # 631352) with *Nde1/Mlu1*. Myc-INPP4B$^{WT}$ or Myc-INPP4B$^{C842A}$[19] were amplified with with *Nde1/Mlu1* restriction sites, then digested and cloned into the *Nde1/Mlu1* site of pLVX-TRE3G-mCherry. All plasmid DNA

sequences were verified by Sanger sequencing (Micromon, Monash University, Australia).

Retroviral transductions were carried out as previously described[64]. For all lentiviral transductions, cells were seeded into a 12-well plate at a density of $4.25 \times 10^4$ cells per well. The next day, fresh media containing 10 μg/mL polybrene and lentiviral or retroviral particles at a multiplicity of infection of 1 was added. Cells were centrifuged at $230 \times g$ for 1 h at room temperature then incubated at 37 °C. After 24 h, the culture media was aspirated and cells were washed three times in

**Fig. 7 Mutant *PIK3CA* promotes increased Wnt gene expression through INPP4B. a** Expression of a panel of 17 Wnt pathway genes was assessed in 1469 ER$^+$ breast cancers from the METABRIC cohort, and stratified by *PIK3CA*-mutation status. The center line indicates the median, the lower bound of the box indicates the 25th percentile, the upper bound of the box represent the 75th percentile, the lower whisker extends from the 25th percentile to the 5th percentile, and the upper whisker extends from the 75th percentile to the 95th percentile. **b–d** MCF-7 cells expressing GFP-INPP4B or GFP-vector were treated with BKM120 (0.5 or 1 µM) or DMSO as a vehicle control for 48 h, then RNA was extracted two-step quantitative RT-PCR was performed using primers for *AXIN2* (**b**), *MYCN* (**c**), or *LEF1* (**d**). Expression was normalized to *RRN18S*. Expression was determined using the ΔΔCt method and expressed relative to vehicle-treated GFP-vector control cells (±SD), which were assigned an arbitrary value of 1 (*n* = 3 experiments). **e** Model of PI3Kα and Wnt/β crosstalk via INPP4B-mediated late endosome formation. In *PIK3CA*-mutant ER$^+$ breast cancers, INPP4B is upregulated and localizes to the late endosome via its interaction with Rab7, which enhances its PI(3,4)P$_2$ 4-phosphatase activity. INPP4B converts PI(3,4)P$_2$ to PI(3)P downstream of PI3Kα, which promotes Hrs-mediated late endosome formation, and increases the lysosomal degradation of GSK3β bound to the β-catenin destruction complex, promoting activation of Wnt/β-catenin signaling. *p* values determined by unpaired two-tailed Mann–Whitney test in **a**, or by one-way ANOVA with Tukey post hoc test in **b**, **c**, **d**.

PBS to remove virus particles before fresh media was added. Cells transduced with lentiviral particles encoding pHIV-1SDmCMV.pre GFP-vector or pHIV-1SDmCMV.pre GFP-INPP4B[6] were selected by fluorescent activated cell sorting (FACS) (Flowcore, Monash University, Australia). Cells transduced with lentiviral particles encoding pLKO.1-puro Non-Mammalian shRNA (Sigma, Cat # SHC002), *INPP4B* #1 mission® shRNA (Sigma, Cat # TRCN0000052721), *INPP4B* #2 mission® shRNA (Sigma, Cat # TRCN0000052722), *HGS* mission® shRNA (Sigma, Cat # TRCN000038-0920), *PIK3C2A* #1 mission® shRNA (Sigma, Cat # TRCN0000002228), or *PIK3C2A* #2 mission® shRNA (Sigma, Cat # TRCN0000002229) lentiviral particles were selected by culturing cells in media containing 1 µg/mL puromycin (Sigma, Cat # P9620) for 1 week, then maintained in media containing 0.5 µg/mL puromycin. Cells co-transduced with lentiviral particles encoding pLVX-EF1α-Tet3G and pLVX-TRE3G-mCherry, pLVX-TRE3G-mCherry-Myc-INPP4B$^{WT}$ or pLVX-TRE3G-mCherry-Myc-INPP4B$^{C842A}$ were selected by culturing in media containing 1 µg/mL puromycin and 600 µg/mL G418 (AG Scientific, Cat # G-1033) for 1 week, then maintained in media containing 0.5 µg/mL puromycin and 300 µg/mL G418. Cells were plated into antibiotic-free media for all experiments. To induce expression of Myc-INPP4B$^{WT}$ or Myc-INPP4B$^{C842A}$, 10 µg/mL doxycycline was added to media when cells were plated.

**Immunoblotting.** Cell lysates were prepared for immunoblotting by washing cells once with TBS on ice followed by direct cell lysis in 40 mM Tris pH 6.8, 4% (w/v) SDS, 20% (v/v) glycerol, 0.0002% (w/v) bromophenol blue, 50 mM DTT. Lysates were boiled for 5 min at 100 °C, and proteins were separated by 10% SDS-PAGE at 150–200 V for 1–1.5 h. Proteins were then transferred to nitrocellulose by electrophoresis at 250 mA for 1.5 h. Immunoblot blocking solution (5% skim milk in TBS) was added to membranes for 1 h at room temperature while rocking. Membranes were incubated in primary antibodies (Supplementary Table 2) diluted in TSB-T overnight at 4 °C while rocking. Membranes were washed three times with TBS-T while rocking for 10 min each. Secondary HRP-conjugated antibodies (Supplementary Table 2) diluted in TBS-T were added for 1 h at room temperature then membranes were washed three times in TBS-T while rocking. Membranes were immersed in ECL Plus for 1 min, then exposed to X-ray film in a dark room and developed using a Fuji processor. Densitometry with ImageQuant version 8.1.1.0 software (GE Healthcare Life Sciences, https://www.gelifesciences.com/) was used to quantify protein bands, with signals being normalized to the housekeeping protein GAPDH.

**Protease protection assay.** Protease protection assays were carried out as previously described[41,42]. Briefly, cells were incubated with 6.5 µg/mL digitonin solution containing 100 mM potassium phosphate pH 6.7, 5 mM MgCl$_2$, 250 mM sucrose for 5 min at room temperature, followed by 30 min on ice. Cells were washed with PBS, and incubated with 1 µg/mL proteinase K (Sigma, Cat # P6556) with or without 0.1% (v/v) Triton X-100 for 10 min. Cells were washed with PBS, and lysed with 40 mM Tris pH 6.8, 4% (w/v) SDS, 20% (v/v) glycerol, 0.0002% (w/v) bromophenol blue, 50 mM DTT, 5 mM PMSF and were boiled for 5 min at 100 °C then immunoblotted.

**Immunofluorescence**

*Indirect immunofluorescence of MCF-10A acini.* Acini structures were fixed and assessed by indirect immunofluorescence as previously described[26]. Acini were fixed with 2% (w/v) paraformaldehyde in PBS for 20 min and permeabilized with 0.05% (v/v) Triton X-100, 130 mM NaCl, 7 mM Na$_2$HPO$_4$, 3.5 mM NaH$_2$PO$_4$, pH 8.0 for 10 min at 4 °C. Acini were washed three times with 100 mM glycine, 130 mM NaCl, 7 mM Na$_2$HPO$_4$, 3.5 mM NaH$_2$PO$_4$, pH 8.0 for 15 min each. Acini were blocked for 1.5 h with primary block containing 10% (v/v) goat serum, 130 mM NaCl, 7 mM Na$_2$HPO$_4$, 3.5 mM NaH$_2$PO$_4$, 7.7 mM NaN$_3$, 0.1% (w/v) BSA, 0.2% (v/v) Triton X-100, 0.05% (v/v) Tween-20, pH 8.0. Acini were blocked for a further 40 min in primary block containing 20 µg/mL goat anti-mouse F(ab)′$_2$ fragment (Jackson ImmunoResearch). Acini were incubated in primary antibodies

(Supplementary Table 2) diluted in primary block overnight at 4 °C, then acini were washed three times with wash buffer (130 mM NaCl, 7 mM Na$_2$HPO$_4$, 3.5 mM NaH$_2$PO$_4$, 7.7 mM NaN$_3$, 0.1% (w/v) BSA, 0.2% (v/v) Triton X-100, 0.05% (v/v) Tween-20, pH 8.0) for 20 min each with gentle rocking. Alexa Fluor® secondary antibodies (Life Technologies), phalloidin (Life Technologies), and/or DAPI (Supplementary Table 2) were added in primary block for 45 min at room temperature. Acini were washed three times with wash buffer for 20 min each with rocking. Acini were washed once with 130 mM NaCl, 7 mM Na$_2$HPO$_4$, 3.5 mM NaH$_2$PO$_4$, pH 8.0, then coverslips mounted with Fluoromount-G (Invitrogen, Cat # 00-4958-02). Images were obtained using a Nikon upright confocal fluorescent microscope (Monash Micro Imaging, Monash University, Australia) and NIS-elements version 4.13 software (Nikon, https://www.microscope.healthcare.nikon.com/), and single z-plane images were taken using the same laser power for all conditions within the same experiment. Images were analyzed using ImageJ version 2.0.0 software (https://imagej.nih.gov)[65].

*Immunofluorescence of endolysosomal structures.* Endosomes and lysosomes were visualized by immunofluorescence using a saponin-based protocol that preserves endosomal structures[66]. Cells were seeded onto 15 mm round coverslips. For inhibitor treatments, the following day media was replaced containing 0.5 µM BKM120 (Selleck Chemicals, Cat # S2247), 1 µM BYL719 (Selleck Chemicals, Cat # S2814), or DMSO as a vehicle control. After 24 h, cells were fixed with 4% (w/v) PFA for 20 min, then washed three times in 50 mM NH$_4$Cl in PBS. Cells were blocked and permeabilized in 2% (w/v) BSA, 0.05% (w/v) saponin in PBS for 1 h. Primary antibodies (Supplementary Table 2) were diluted in block, and added overnight at 4 °C then cells were washed three times in PBS. Alexa Fluor® secondary antibodies, phalloidin, and DAPI (Supplementary Table 2) were diluted in block, and added for 1 h at room temperature. Cells were washed three times with PBS, and mounted onto slides with Fluoromount-G. Slides were imaged using a Leica SP8 invert confocal laser scanning microscope (Monash Micro Imaging, Monash University, Australia) and LAS X version 3.5.6.21594 software (Leica, https://www.leica-microsystems.com/), and single z-plane images were taken using the same laser power for all conditions within the same experiment.

To visualize INPP4B at endosomes, cells were pretreated with saponin before fixation to remove cytoplasmic proteins and retain proteins bound by intracellular membranes as previously described[67]. Media was aspirated from cells, then cells were permeabilized with 0.02% (w/v) saponin, 25 mM KCl, 2.5 mM MgCl$_2$, 25 mM HEPES, pH 7.4 for 30 s. Cells were then fixed and stained using the saponin-based protocol as described above.

For *PIK3CA* siRNA depletion, GFP-vector and GFP-INPP4B expressing MCF-7 cells were reverse transfected with *PIK3CA* siRNA #1 (Qiagen, Cat # SI02622207), *PIK3CA* siRNA #2 (Qiagen, Cat # SI02665369), or NT siRNA (Qiagen, Cat # 1027381) using Lipofectamine RNAiMAX (Invitrogen, Cat # 13778075) according to the manufacturer's directions (Transfecting Stealth™ RNAi or siRNA into MCF7 Cells Using Lipofectamine™ RNAiMAX). Cells were then incubated for 48 h at 37 °C before fixation and immunostaining was performed.

*Detection of PI(3)P by immunofluorescence.* To visualize intracellular PI(3)P by immunofluorescence, used the "Golgi" staining method was used[68]. Cells were seeded onto 15 mm round coverslips. The following day, cells were fixed in 2% PFA (w/v) for 15 min. Cells were washed three times with 50 mM NH$_4$Cl in PBS, and then permeabilized with Buffer A (20 mM PIPES, 137 mM NaCl, 2.7 mM KCl, pH 6.8) with 20 µM digitonin for 5 min. Cells were washed three times in Buffer A and then blocked with 5% (v/v) goat serum, 50 mM NH$_4$Cl in Buffer A for 45 min. About 8 µg/mL recombinant GST-2xFYVE$^{Hrs}$[69] was diluted with 5% (v/v) goat serum in Buffer A and added to cells for 30 min. Cells were washed twice with Buffer A then primary antibodies (Supplementary Table 2) were added with 5% (v/v) goat serum in Buffer A for 1 h. Cells were washed twice with Buffer A. Alexa Fluor® secondary antibodies, phalloidin, and DAPI (Supplementary Table 2) were diluted with 5% (v/v) goat serum in Buffer A and added to cells for 45 min. Cells were washed four times with Buffer A. Cells were post-fixed in 2% (w/v) PFA then washed three times in 50 mM NH$_4$Cl in PBS and once with dH$_2$O. Coverslips were mounted

onto slides with Fluoromount-G and single z-plane images were obtained using a Leica SP8 invert confocal laser scanning microscope (Monash Micro Imaging, Monash University, Australia) and LAS X version 3.5.6.21594 software.

*DQ Red BSA assay.* Cells were seeded onto 15 mm round coverslips. The following day, cells were treated with 40 µg/mL DQ™ Red BSA (Life Technologies, Cat # D12051) for 5 h. Cells were washed three times in PBS, then fixed in 4% (w/v) PFA for 20 min Cells were washed three times in PBS, then DAPI (1 µg/mL) diluted in PBS was added for 30 min. Cells were washed three times in PBS. Coverslips were mounted onto slides with Fluoromount-G and single z-plane images were obtained using a Leica SP8 invert confocal laser scanning microscope (Monash Micro Imaging, Monash University, Australia) and LAS X version 3.5.6.21594 software.

*Particle analysis.* Images were analyzed using ImageJ version 2.0.0 software. For particle analysis, the same channel threshold was applied to all images from the same experiment. The "analyse particle" plugin was used to determine the number of EEA1-positive, CD63-positive, LAMP1-positive, 2xFYVE-positive, or BSA-DQ-positive puncta greater than 0.1 µm$^2$ per cell, or the number of GSK3β-positive puncta greater than 0.2 µm$^2$ per cell.

Particle co-localization analysis was performed as previously described[70]. The CD63-positive or EEA1-positive puncta were used to construct a mask that was overlaid with the 2xFYVE-positive puncta, and the number of double positive puncta with >30% overlap in area were quantified.

*Detection of GSK3 at endosomes.* Cells were seeded onto 15 mm round coverslips and cultured overnight, then cells were transfected with RFP-GSK3 (Addgene, Cat # #29679[40]) using Lipofectamine 3000 (Invitrogen, Cat # L3000-015). Forty-eight hours later, cells were incubated with 50 nM LysoTracker Deep Red (Invitrogen, Cat # L12492) for 1 h. Cells were fixed in 4% (w/v) PFA for 20 min, and washed three times with PBS. DAPI (1 µg/mL) was diluted in PBS and added for 30 min. Cells were washed three times in PBS, then mounted onto slides with Fluoromount-G. Slides were imaged using a Leica SP8 invert confocal laser scanning microscope (Monash Micro Imaging, Monash University, Australia) and LAS X version 3.5.6.21594 software, and single z-plane images were taken using the same laser power for all conditions within the same experiment.

### BrdU uptake assay

BrdU uptake assays were performed using the 5-Bromo-2′-deoxy-uridine Labeling and Detection Kit I (Roche, Cat # 11296736001) according to the manufacturer's instructions. Briefly, cells were plated onto 15 mm coverslips in a 12 well plate at a density of ~7.5 × 10$^4$ cells per well. After 24 h, cells were serum starved either overnight (MCF-7 cells) or for 48 h (MCF-10A cells) to synchronize the cell cycle at G0/G1 phase. Cells were incubated with 10 µM BrdU labeling solution diluted in serum free media either overnight (MCF-7) or for 2 h (MCF-10A). For inhibitor treatments, cells were serum-starved and BrdU-treated in the presence of IWP-2 (Sigma, Cat # I0536), LGK-974 (Selleck Chemicals, Cat # S7143), IWR-1-endo (Merck, Cat # 681669), or DMSO as vehicle control. Cells were washed with PBS three times, then fixed using 50 mM glycine, 70% (v/v) ethanol at −20 °C for 20 min. Cells were washed three times in PBS, and BrdU primary antibody was added for 30 min at 37 °C. Cells were washed in PBS three times, and incubated with anti-mouse-Ig-fluorescein or Alexa Fluor® secondary antibodies for 30 min at 37 °C. Cells were washed three times in PBS, then incubated with propidium iodide (1 µg/mL) or DAPI (1 µg/mL) in PBS for 15 min at room temperature. Following a final wash, coverslips were mounted onto glass slides with Fluoromount-G and imaged using a Nikon upright fluorescent confocal microscope (Monash Micro Imaging, Monash University, Australia) and NIS-elements version 4.13 software. Images were analyzed using ImageJ version 2.0.0 software. The percentage of BrdU-positive cells was determined using the "cell counter" plugin.

### Electron microscopy

*Immuno-gold double labeled transmission electron microscopy.* Tokuyasu sample preparation and immuno-gold labeled transmission electron microscopy was performed as described[71]. Cell monolayers were cultured in 10 cm plates and allowed to reach 70% confluency. Cells were then fixed overnight at 4 °C with 0.1 M phosphate buffered 2% (w/v) paraformaldehyde and 0.2% (w/v) glutaraldehyde. The fixed samples were scraped and pelleted in 12% (w/v) gelatin in 0.1 M phosphate buffer at 37 °C, which was allowed to set at 4 °C before being cut into small cubes measuring ~0.5 mm on each edge. The gelatin embedded cells were infiltrated with 2.3 M sucrose in 0.1 M phosphate buffer at 4 °C overnight on a rocker. The sucrose infiltrated gelatin blocks were mounted on cryo-pins then frozen in liquid nitrogen for cryo-ultramicrotomy. Frozen samples were trimmed at −100 °C and sectioned at −120 °C using a Cryo-EM UC7 ultramicrotome (Leica Microsystems) equipped with a 45° diamond cryo-trimming knife (Diatome) and a 35° diamond cryo-immuno knife (Diatome). Cryo-sections were retrieved by pick-up loop with a droplet of phosphate buffered 1% (w/v) methyl cellulose and 1.15 M sucrose, then deposited on carbon-coated formvar grids for immunolabeling. The grids were prepared for immunolabeling by melting upside down on PBS at 37 °C, then rinsing the grids on droplets of 0.02 M glycine in PBS four times for 2 min each. The grids were blocked with 1% (w/v) BSA in PBS for 5 min, incubated with

mouse anti-CD63 primary antibody (Supplementary Table 2) in 1% (w/v) BSA in PBS (room temperature; 1 h), then rinsed with 0.1% (w/v) BSA in PBS (5 × 2 min). The grids were then incubated with a bridging antibody (rabbit anti-mouse IgG) (Supplementary Table 2) diluted in 1% (w/v) BSA in PBS (room temperature; 20 min), and rinsed (5 × 2 min) with 0.1% (w/v) BSA in PBS. The grids were then incubated with Protein-A conjugated 15 nm gold particles (1:50, Cell Biology, Utrecht) diluted in 1% (w/v) BSA in PBS (room temperature; 20 min) before rinsing with PBS (5 × 2 min). After stabilization of the reaction by 1% glutaraldehyde (w/v) in PBS for 5 min the grids were rinsed on droplets of 0.02 M glycine in PBS (4 × 2 min) and were blocked with 1% (w/v) BSA in PBS (5 min). Grids were incubated with goat anti-GFP-Biotin (Supplementary Table 2) in 1% (w/v) BSA in PBS (room temperature; 1 h) and incubated with a bridging rabbit anti-biotin antibody (Supplementary Table 2) then rinsed with 0.1% (w/v) BSA in PBS (5 × 2 min). The grids were then incubated with Protein-A conjugated 10 nm gold particles diluted in 1% (w/v) BSA in PBS (room temperature; 20 min) and rinsed with PBS (5 × 2 min). After stabilization of the reaction by 1% glutaraldehyde (w/v) in PBS for 5 min, grids were rinsed in distilled water (8 × 2 min). Finally, the grids were stained with 2% (w/v) uranyloxalate (pH 7; room temperature; 5 min) and 0.4% (w/v) uranyl acetate in 1.8% (w/v) aqueous methyl cellulose (pH 4; 4 °C; 5 min), and dried in a thin film of the final stain in the center of a wire loop. EM imaging was performed on a Jeol1400-Flash TEM and a FEI Tecnai 12 TEM (Monash Ramaciotti Centre for Cryo Electron Microscopy, Monash University, Australia).

*BSA-gold internalization.* BSA-gold uptake studies were performed as described previously[72]. Cells were washed once with PBS then incubated for 1 h with phenol red-free DMEM without FCS. Cells were incubated with BSA-Au 5 or 10 nm (OD = 5) for 3 h at 37 °C. Standard fixation was performed in 2.5% (v/v) glutaraldehyde in 0.1 M sodium-cacodylate buffer. Cells were post-fixed in reduced osmium tetroxide, dehydrated and embedded in Epon 812 resin. 75 nm sections were placed on a grid coated with formvar film and carbon. EM imaging was performed on a Jeol1400-Flash TEM (Monash Ramaciotti Centre for Cryo Electron Microscopy, Monash University, Australia).

### Anchorage-independent cell growth assays

0.7% (w/v) agar underlays were prepared in a 6-well plate by combining 2 mL 5% (w/v) agar stock with 12 mL media, and plating 2 mL of agar/media per well. Underlays were incubated for 30 min at room temperature to solidify. 0.3% (w/v) agar/cell suspension overlays were prepared by adding 1 mL agar to 13 mL media, then transferring 12 mL agar/media solution to 3 mL growth media containing 6000 cells. About 5 mL of agar/cell suspension was added to each of three underlays, and cells were incubated at 37 °C. For inhibitor treatments, 2 µM IWP-2 (Sigma) or DMSO as a vehicle control was added to agar/cell suspension before plating. After 4 weeks, colonies were imaged using a Leica DFC295 camera mounted to a Leica M165C microscope. Colony numbers and size were quantified using ImageJ version 2.0.0 software.

### Whole proteome analysis by LC-MS/MS

*Protein preparation.* Cells were cultured in growth media, then lysed in proteome profiling buffer (50 mM HEPES-NaOH, 150 mM NaCl, 0.5% (v/v) Triton X- 100, 1 mM EDTA, 1 mM EGTA, 10 µg/mL aprotinin,10µg/mL leupeptin, 1 mM PMSF, 10 mM NaF, 50 ng/mL calyculin A, 1% (v/v) phosphatase inhibitor mixture 2, 2.5 mM Na$_3$VO$_4$, pH 7.5). Total protein measurements were determined using the Bicinchoninic acid protein assay (Bio-Rad). About 100 µg of protein extracts were denatured with 6 M urea in 25 mM ammonium bicarbonate, before reduction with 5 mM TCEP at 37 °C for 1 h and alkylation with 32 mM iodoacetamide in the dark for 1 h. Alkylation was stopped by addition of 27 mM DTT. The samples were diluted 1:10 with ammonium bicarbonate and digested with a 1:50 ratio of modified trypsin (Promega) to protein weight at 37 °C for 18 h. Tryptic digests were slightly acidified with 10% (v/v) TFA to pH 2–3, desalted with a C18 spin column (ThermoFisher Scientific), and eluted with 0.1% (v/v) TFA/40% (v/v) acetonitrile. Peptides were dried with a speed vacuum and resuspended in 2% (v/v) acetonitrile/ 0.1% (v/v) FA prior to mass spectrometry analysis.

*Mass spectrometry analysis.* Samples were analyzed on an UltiMate 3000 RSLC nano LC system (ThermoFisher Scientific) coupled to an LTQ-Orbitrap mass spectrometer (LTQ-Orbitrap, ThermoFisher Scientific). Peptides for analysis were loaded via an Acclaim PepMap 100 trap column (100 µm x 2 cm, nanoViper, C18, 5 µm, 100 Å, ThermoFisher Scientific) and subsequent peptide separation was on an Acclaim PepMap RSLC analytical column (75 µm x 50 cm, nanoViper, C18, 2 µm, 100 Å, ThermoFisher Scientific). For each liquid chromatography-tandem mass spectrometry (LC-MS/MS) analysis, an estimated amount of 1 µg of peptides determined by nanodrop analysis was loaded on the precolumn with microliter pickup. Peptides were eluted using a 2 h linear gradient of 80% (v/v) acetonitrile/ 0.1% (v/v) FA gradient flowing at 250 nL/min using a mobile phase gradient of 2.5–42.5% (v/v) acetonitrile. The eluting peptides were interrogated with an Orbitrap mass spectrometer. The data-independent acquisition (DIA) method consisted of a survey scan (MS1) at 35,000 resolution (automatic gain control target 5e6 and maximum injection time of 120 ms) from 400 to 1220 m/z followed by tandem MS/MS scans (MS2) through 19 overlapping DIA windows increasing

from 30 to 222 Da. MS/MS scans were acquired at 35,000 resolution (automatic gain control target 3e6 and auto for injection time). Stepped collision energy was 22.5%, 25%, 27.5%, and a 30 m/z isolation window. The spectra were recorded in profile type.

*Spectral library generation.* Samples were decomplexed with HpH fractionation and data-dependent acquisition (DDA) measurements of five fractions of each sample were performed. The DDA spectra were analyzed with the MaxQuant version 1.5.2.8 analysis software (http://www.maxquant.org./)[73] using default settings. Enzyme specificity was set to Trypsin/P, minimal peptide length of 6, and up to three missed cleavages were allowed. Search criteria included carbamidomethylation of cysteine as a fixed modification, oxidation of methionine and acetyl (protein N terminus) as variable modifications. The mass tolerance for the precursor was 4.5 ppm and for the fragment ions was 20 ppm. The DDA files were searched against the human UniProt fasta database (version 2015-08, 20,210 entries) and the Biognosys HRM calibration peptides. The identifications were filtered to satisfy FDR of 1% on peptide and protein level. The spectral library was generated in Spectronaut version 8 software (Biognosys, https://biognosys.com/) and normalized to iRT peptides[74]. A peptide identification required at least three transitions in quantification. Quantification was based on the top three proteotypic peptides for each protein[75] and exported as an Excel file with Spectronaut version 8 software[74].

*HRM-DIA data analysis.* The DIA data were analyzed with Spectronaut version 8. The default settings were used for the Spectronaut search. Retention time prediction type was set to dynamic iRT. Decoy generation was set to scrambled with no decoy limit. Interference correction on MS2 level was enabled. The false discovery rate (FDR) was set to 1% at peptide level. Student $t$-test of cell lines was performed to determine differential proteins ($p$ value <0.05).

*Functional analysis.* Functional annotation of the proteome was conducted using database for annotation, visualization, and integrated discovery (DAVID) version 6.7 (https://david.ncifcrf.gov/)[76]. Overrepresented functional categories among proteins enriched in each sample population were relative to a background of all identified proteins in the study. Criteria for reported functional enrichment required a fold enrichment >1.5, FDR < 5 and $p$ value <0.05.

**Immunoprecipitations.** Immunoprecipitations of HA tagged proteins were carried out as previously described[77]. Cells were cultured in 10 cm plates until 70% confluency was reached. Cells were washed once with ice-cold wash buffer (TBS containing 10 mM NaF, 10 mM β-glycerophosphate, 1 mM $Na_3VO_4$, protease inhibitor cocktail (Roche, Cat # 11836153001), pH 7.4). Cells were scraped in 500 μL of wash buffer containing 1% (v/v) Triton X-100 into tubes on ice, and rocked for 1 h. Samples were centrifuged at 14,000 × $g$ for 5 min at 4 °C. An aliquot of the soluble lysate fraction was kept to run on SDS-PAGE. A total of 2.5 μL HA antibody (Supplementary Table 2) was added to the soluble lysate fraction, and tubes were incubated with rocking for 1 h at 4 °C. About 60 μL of protein A-Sepharose (Invitrogen, Cat # 101042) was added and tubes were rocked overnight at 4 °C. Samples were centrifuged at 14,000 × $g$ for 1 min at 4 °C, and supernatant was discarded. Pellets were washed three times in ice-cold wash buffer. Proteins were eluted for immunoblotting analysis by adding SDS-reducing buffer, and boiling for 5 mins.

Immunoprecipitations of GFP-tagged proteins were performed using GFP-Trap (Chromotek, Cat # gta-20) according to the manufacturer's instructions. Cells were cultured in 10 cm plates until 70% confluency was reached. Cells were washed once with ice-cold dilution buffer (10 mM Tris, 150 mM NaCl, 0.5 mM EDTA, 10 mM NaF, 10 mM β-glycerophosphate, 1 mM $Na_3VO_4$, protease inhibitor cocktail, pH 7.5) and then scraped in 300 μL dilution buffer with 0.5% (v/v) NP-40 into tubes on ice. Tubes were left on ice for 30 min with extensive pipetting every 10 min. Lysates were centrifuged at 17,000 × $g$ for 10 min at 4 °C. An aliquot of the soluble lysate fraction was kept to run on SDS-PAGE, and the remainder was made up to 1 mL with dilution buffer. GFP-Trap agarose beads were prewashed three times with dilution buffer, and the diluted cell lysates were resuspended with GFP-Trap beads for 1 h at 4 °C while rocking. Samples were centrifuged at 2000 × $g$ for 2 min at 4 °C, and supernatant was discarded. Pellets were washed three times with dilution buffer. For immunoblotting analysis, proteins were eluted from beads by adding SDS reducing buffer, and boiling for 5 min. For mass spectrometry analysis, proteins were eluted using 0.2 M glycine, pH 2.5 followed by neutralization with 1 M Tris, pH 8.

**In solution digestion of immunopurified protein complexes and LC-MS/MS analysis.** Disulfide bridges were reduced for 30 min at 37 °C using 5 mM TCEP (Thermo Scientific) and free cysteine residues were alkylated for 30 min at room temperature in the dark using 20 mM iodoacetamide (Sigma). Trypsin (Sigma) was used to digest the proteins into peptides and the reaction was stopped after 16 h by the addition of formic acid to 1% (v/v). All samples were desalted using P-10 ZipTip columns (Agilent, OMIX-Mini Bed 96 C18), vacuum-dried and reconstituted in buffer A (0.1% (v/v) formic acid, 2% (v/v) acetonitrile) prior to mass spectrometry. Using a Dionex UltiMate 3000 RSLCnano system equipped with a Dionex UltiMate 3000 RS autosampler, an Acclaim PepMap RSLC analytical

column (75 μm x 50 cm, nanoViper, C18, 2 μm, 100 Å; ThermoFisher Scientific) and an Acclaim PepMap 100 trap column (100 μm x 2 cm, nanoViper, C18, 5 μm, 100 Å; ThermoFisher Scientific), the tryptic peptides were separated by increasing concentrations of 80% (v/v) acetonitrile/0.1% (v/v) formic acid at a flow of 250 nL/min for 60 min and analyzed with an Orbitrap Fusion Tribrid mass spectrometer (ThermoFisher Scientific) using in-house optimized parameters to maximize the number of peptide identifications. The acquired raw files were searched against a human SwissProt database using Byonic version 3.1.0 software (Protein Metrics, https://www.proteinmetrics.com/) and/or MaxQuant version 1.5.2.8 software to obtain peptide sequence or quantitative information. Only peptides identified at a FDR of 1% based on a decoy database were considered for further analysis.

**Malachite green phosphatase assay.** INPP4B phosphatase assays were performed using the Malachite Green Phosphatase Assay (Echelon, Cat # K-1500) according to the manufacturer's instructions. Recombinant purified INPP4B protein (Andrew Ellisdon) was preincubated with recombinant purified Rab7 protein (Abcam, Cat # 103507) for 30 min gently shaking at 37 °C in phosphatase buffer. Samples were placed on ice for 2 min. About 50 μM $PI(3,4)P_2$ diC8 (Echelon, Cat # P-3408) was added and reactions were incubated at 37 °C for 20 min. Samples were placed on ice, and 20 μL of each sample was loaded into a 96-well plate. A total of 80 μL of malachite green reagent was added and after 15 min, the color development was measured at 655 nm using a PheraStar plate reader (BMG Labtech). Phosphate standards were used to generate a four-parameter non-linear regression standard curve to calculate phosphate release.

**RNA analysis.** RNA was extracted from cells using the Isolate II RNA extraction kit (Bioline, Cat # BIO-52073) according to the manufacturer's instructions. RNA concentration was determined using a Nanodrop 1000 spectrophotometer (ThermoFisher Scientific). Extracted RNA was diluted to 50 ng/mL, and subjected to two-step qRT-PCR using the iScript gDNA clear cDNA synthesis kit (Bio-Rad, Cat # 172-5035) and the QuantiTect SYBR Green PCR Kit (Qiagen, Cat # 204143) according to the manufacturer's instructions. Reactions were subjected to thermocycling using a CFX384 Real Time PCR System (Bio-Rad), and analyzed using CFX Manager version 3.1 software (Bio-Rad, https://www.bio-rad.com). All primers used are listed in Supplementary Table 1. A no template control without cDNA and a no reverse transcriptase control were included to ensure no genomic DNA was present. The relative expression of the gene of interest was compared to the standard housekeeping gene *GAPDH* or *RRN18S* and quantified using the ΔΔCT method[61].

**nCounter® PanCancer Pathways Panel.** The nCounter® PanCancer Pathways Panel (nanoString, Cat # XT-CSO-PATH1-12) was used to screen for differential expression of 770 mRNAs, including 730 genes of interest and 40 reference genes. About 70 μL of hybridization buffer was added to the Reporter CodeSet then 8 μL of this was added to 2 tubes. A total of 40 ng of extracted RNA was added to the CodeSet and RNase-free water was added to a final volume of 13 μL. Two microliters of Capture ProbeSet was added to each tube and reagents were mixed before incubation at 65 °C for 18 h. Following incubation, 17 μL $H_2O$ was added to each sample. Thirty microliters of the samples were loaded into the nanoString cartridge, which was then loaded into the nanoString nCounter Sprint Profiler machine and the data acquisition program run. Data were analyzed using nSolver version 3.0 software (nanoString, https://www.nanostring.com/). The internal positive controls in the PanCancer Pathways codeset were above the limit of detection and quality control analysis confirmed overall assay efficiency and linearity. Counts from the 730 genes of interest were normalized to positive controls and to a panel of 15 reference genes to correct for differences in sample input. The reference genes were selected as the top 15 genes with the lowest %CV (coefficient of variation) and counts of >100. Background threshold of detection was determined by calculating the mean plus two standard deviations of internal negative controls containing nonspecific target probes. Genes below this FDR were not considered in the analysis.

**Xenograft tumor growth.** MCF-7 cells stably co-expressing GFP-alone;NT shRNA, GFP-alone;*Hrs* shRNA, GFP-INPP4B;NT shRNA or GFP-INPP4B;*Hrs* shRNA were maintained in 15 cm plates at 70% confluence in culture media. On the day of inoculation, cells were washed twice with PBS and detached from the plates in 5 mL versene at 37 °C for 20–30 min. Five milliliter of culture media was added to the flasks, then the cell suspension was centrifuged at 1000x$g$ for 3 min. The cell pellet was resuspended in 50 mL PBS and the total cell number was determined using a haemocytometer. $1 × 10^6$ cells/mouse were resuspended in 7.5 μL PBS, then 7.5 μL of growth factor reduced Matrigel (Corning, Cat # 356231) was added to the cell suspension. Seven- to eight-week-old female athymic BALB/c-Fox1nu/Asb mice (Australian BioResources) were anaesthetized with isoflurane (1–2% in oxygen) then 15 μL of cell-Matrigel mixture containing $1 × 10^6$ cells was injected into the right fourth mammary fat pad of eight mice/cell line. Estrogen pellets (17β estradiol 90 day release, 1.5 mg/pellet, Innovative Research of America, Cat # NE-121) were implanted subcutaneously on the back of the neck. Two weeks post-injection, the width and length of palpable tumors were measured with digital callipers. Measurements were repeated three times per week for 2 weeks. Tumor

volume was calculated using the equation (length) x (width)$^2$/2. Once tumor size approached the ethical endpoint of 1000 cm$^3$, mice were humanely killed and tumors were excised and the weight of each tumor was recorded. Mice that died for unknown reasons were excluded from the analysis (one mouse from GFP-INPP4B; NT shRNA group, and one mouse from GFP-INPP4B;*Hrs* shRNA group). All studies involving BALB/c-Fox1nu/Ausb mice were undertaken after receiving ethical approval from the Animal Ethics Committee, Monash University, Australia (project number MARP/2017/168). Mice were housed at 18–24 °C with 40–70% humidity on a 12-hour light-dark cycle, with access to food and water ad libitum (Animal Research Laboratory, Monash University, Australia). Mice were randomly assigned to eight per cage and each cage was used as an experimental group. Mouse numbers were determined by previous experience in xenograft studies[6].

**Statistics and reproducibility**. Genetically modified cells in each experiments were derived from the same pool of parent cells. Cells were randomly assigned to treatment or control groups. No sample size calculations were performed. For electron microscopy experiments, at least 50 cell profiles were qualitatively assessed in a single experiment. Immunoprecipitation-mass spectrometry analysis was performed using two independent replicates as this was sufficient to identify Rab7 binding, which was validated by immunoblotting. All other experiments were repeated at least three times independently to ensure statistical significance of the results. Statistical analysis was performed using Prism version 7.0 (GraphPad, https://www.graphpad.com). Details of statistical testing can be found in the figure legends. Differences between groups were considered statistically different for *p* values <0.05.

**Reporting summary**. Further information on research design is available in the Nature Research Reporting Summary linked to this article.

## Data availability

The mass spectrometry proteomic.htrms files,.RAW files, and result files have been deposited to the Mass spectrometry Interactive Virtual Environment (MassIVE) consortium (https://massive.ucsd.edu/ProteoSAFe/dataset.jsp?task=a8b8b555fc954bd69e1f9e9e7dd337aa) with dataset identifier: MSV000085526. Data can also be found in the Proteome Xchange (http://proteomecentral.proteomexchange.org/cgi/GetDataset?ID=PXD019503) under the Pride identifier: PXD019503. The DDA files were searched against the human UniProt fasta database (version 2015-08, 20,210 entries, https://ftp.uniprot.org/pub/databases/uniprot/previous_releases/release-2015_08/). Source data and uncropped blots are provided with this paper. All other data that support the findings of this study are available from the corresponding author upon reasonable request. Source data are provided with this paper.

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

## Acknowledgements

This work was supported by an NHMRC grant APP1104614. This study utilized the Monash Micro Imaging, Monash Histology Platform, Flowcore, Monash Proteomics & Metabolomics Facility, Monash Animal Research Platform, Micromon, Monash Ramaciotti Centre for Cryo Electron Microscopy, Monash University, Australia. The authors thank Joan Clark (Monash Ramaciotti Centre for Cryo Electron Microscopy) for excellent technical assistance.

## Author contributions

Conceptualization ideas, C.A. Mitchell., C.A. McLean., L.M.O., and S.J.R.; Methodology, N.R., R.G., M.J.E., A.S., F.C., S.L., C.G.F., G.K., H.E.A., G.R., A.P., A.M.E., and R.J.D.; Investigation, S.J.R., L.M.O., V.M.J.O., R.B.S., E.V.N., and S.A.H.; Resources, V.M.J.O., R.B.S., A.P., H.E.A., A.M.E., and C.A. McLean.; Writing—Original draft preparation, C.A. Mitchell, L.M.O., and S.J.R.; Writing—Review & editing preparation, all authors; Supervision, C.A. Mitchell., L.M.O., and S.J.R.; Funding acquisition, C.A. Mitchell., C.A.McLean., and A.M.E.

## Competing interests

C.A. Mitchell., S.J.R., and L.M.O. are inventors on a planned patent application relating to the use of Wnt inhibitors for breast cancers with increased INPP4B expression. The remaining authors declare no competing interests.
