## [Peer Review File · Nature Communications]

REVIEWER COMMENTS

Reviewer #1 (Remarks to the Author):

In this manuscript, Rodgers et al. report that INPP4B exerts an oncogenic function in promoting ER+ breast cancer growth through enhancing late endosome formation/Wnt-b-catenin signaling activation. Through data mining the authors observed that increased INPP4B mRNA and protein levels in ER+ breast cancer patients. To examine INPP4B function in regulating ER+ breast cancer growth, the authors overexpressed INPP4B in ER+ breast cancer lines MCF7 and T47D. As a result, overexpression of INPP4B facilitated cell growth albeit reducing AKT activation. Further the authors showed that INPP4B expression facilitated PI(3)P accumulation and late endosome formation, and blocking late endosome function by Hrs depletion partially rescued INPP4B overexpression induced growth advantages. Moreover, the authors found INPP4B facilitated Wnt/b-catenin activation which is also necessary for INPP4B expression induced growth increase. Overall, the authors reveal an interesting oncogenic function for INPP4B in PIK3CA mutated ER+ breast cancer through controlling PI(3)P/late endosome/Wnt signaling. The manuscript is well written and experiments were designed in a rational manner to support major conclusions. However, this study heavily relies on data obtained from INPP4B overexpression in an ER+ breast cancer line already with high INPP4B expression that may be artificial. In addition, a lack of causal relationship for the proposed signaling reduces the enthusiasm from the reviewer.

Major concerns:

1. Instead of relying on INPP4B overexpressing MCF-7 and T47D lines, the authors need to obtain more evidence from endogenous INPP4B depleted MCF-7 or T47D or other ER+ breast cancer lines.
2. The causal relationship for the proposed INPP4B/PI(3)P/late endosome formation/Wnt signaling would need to be established. For example, how PI(3)P/late endosome regulates Wnt signaling?
3. This study focuses on PIK3CA mutated ER+ breast cancer settings, but some key questions are not revealed, including (1) Is ER necessary to prime INPP4B function in this setting? (2) Is PIK3CA oncogenic mutant necessary to prime INPP4B function in this setting?

Minor concerns:

1. Figure 1: INPP4B DNA alternation should also be examined using these datasets, especially in ER+ group;
2. Line 136: should be Figure 2D and 2E.
3. Figure 2D/2E: growth between normal serum condition and serum deprivation condition should be compared.
4. Figure 2F: AKT- if this is total AKT or AKT1 should be clarified. Reduction of AKT phosphorylation upon ectopic INPP4B expression is weak. INPP4B/GFP blot is needed in this panel to reveal INPP4B expression levels.
5. Figure 2G-2H: normalization should be done using time 0 (starvation) condition as the basal line to reveal dynamic AKT phosphorylation changes upon EGF stimulation. In addition, a catalytic dead INPP4B is needed here as an additional negative control.
6. Figure 2 is largely depending on overexpression of INPP4B- given as the authors claimed earlier that increased INPP4B mRNA/protein expression is observed in ER+ breast cancer, what is the rationale here to rely on further INPP4B overexpression for these studies? Why not just examine what happens upon knockdown of endogenous INPP4B?
7. Figure S2F: Sample loadings are not even.
8. Figure S2G: An increased PTEN protein level is observed in GFP-INPP4B sample- this is not consistent with the authors' claim.
9. Supple Figure 2B: what is the band between EV and INPP4BWT lane?
10. Figure 2L: Ki67 should be normalized to DAPI stain. In contrary to Figure 2I, no obvious increased in the size of acini, nor increased number of DAPI stain is observed in Figure 2L upon expression of WT-INPP4B.

11. Figure 3B: for IP-mass spec experiments, the rationale for starvation/EGF stimulation would need to be provided.
12. Figure 3C: An interaction between INPP4B and Rab7A would need to be confirmed at endogenous levels.
13. Figure S3A: the input signals are not even, thus the interpretation of binding differences is not reliable.
14. Figure 4: the authors nicely showed that overexpression of INPP4B facilitates late endosome formation. Is this accompanied by increased levels of cellular PI(3)P? Does blocking PI(3)P prevents INPP4B overexpression induced late endosome formation?
15. Figure S5C/S5D: genetic depletion of PIK3CA is needed to validate inhibitor effects on late endosome formation.
16. Figure 5A: shHrs reduces CD63+ late endosome- but it seems this late endosome reduction does not affect 3D growth or tumor growth. Figure 5 reveals a critical function for Hrs in mediating INPP4B overexpression induced growth advantage, but if PI(3)P or late endosome formation plays a role here, or if this is mediated by Hrs function on late endosome is not well established.
17. Figure 6D: increases of non-phospho-b-catenin are quite minor in Wnt3a treated samples. In addition, in GFP-INPP4B expressing cells upon double treatment there is no further reduction of GSK3b but why there is a further increase in non-phospho-b-catenin signal? Does this suggest this change in non-phospho-b-catenin is not mediated by GSK3b protein level changes?
18. Figure 6I: can effects of pharmacological inhibitors be decapitated by genetic manipulation of b-catenin signaling?

Reviewer #2 (Remarks to the Author):

The authors report on how a phosphatase that converts PI(3,4)P₂ into PI(3)P, INPP4B, contributes to breast cancer through membrane trafficking that causes increased Wnt signaling. The authors provide a compelling and thorough analysis of INPP4B from the level of cell signaling to in vivo using mouse models. This discovery is important as it offers novel insights into the field of cancer cell signaling downstream of PI3K and PTEN. The new findings on the oncogenic activity of INPP4B offer a mechanism to understand how INPP4B contributes to the progression of ER+ breast cancer. Previous work demonstrated that INPP4B was inactivated in triple negative breast cancer. Conversely, overexpression of INPP4B had been reported to increase tumorigenesis on other cancers. Here, the authors report that in the case of ER+ breast cancers, expression of INPP4B is increased at the level of both mRNA and protein (pg. 5). Authors go on to show that INPP4B promotes proliferation and growth, even though AKT signaling is inhibited (pg. 6). Mechanistically, INPP4B interacts with Rab7 at late endosomes and subsequently increases endosomal PI3K conversion of PI(3,4)P₂ to PI(3)P.

Main Comments

- 1) The conclusions of the work are nicely summarized in a schematic depiction that is currently presented as the last figure of the supplemental text. It is essential that this model be included as part of the final figure in the main text. It tells the entire tale and will greatly help the readership. Other data can be accommodated in the supplemental material to make space for this explanatory diagram.
- 2) Given the recent emerging roles of lysosomal catabolism and Wnt-induced macropinocytosis in promoting cancer cell metabolism, the authors could include discussion about the possible physiological implications of lysosomal activity in ER+ breast cancer in relation to other cancer types. They convincingly show that INPP4B activates the canonical Wnt pathway through increased lysosomal degradation of GSK3, increases total number of lysosomes, and endocytosis and

degradation of extracellular serum protein using BSA-DQ. Authors should therefore discuss the general implications of late endosome formation on cell signaling in cancer.

Minor Comments

1) The authors clearly demonstrate that the oncogenic roles of INPP4B occur upstream of Wnt signaling through endolysosomal signaling, as Wnt inhibition or HRS knock down was sufficient to block tumor growth. HRS depletion studies were performed in cultured cells, agar colony assay, and in xenograft mice models. They show that GSK3 is degraded in lysosomes on page 11 (Figure 6 and Supplementary Figure 7) by protein levels in the presence of GFP-INPP4B expression, but not at the level of mRNA. Further, GSK3 levels were rescued with HRS knockdown. This supports a model whereby GSK3 is being degraded in lysosomes versus proteasomes. We suggest that including the ratios of GSK3 protein expression relative to GAPDH under the western blot lanes of Figure 6D could help readers better appreciate the decrease in GSK3 protein levels.

2) Figure 6G makes the very important point that INPP4B increases sequestration of GSK3 in late endosomes. While the data as is makes the point, it is not very impressive. Digitonin usually gives better results than saponin. The authors could consider either co-staining with a late endosomal marker such as CD63, or the more exacting protease protection in live cells described in Albrecht et al PNAS 2018. Figure 6G should be improved.

3) On page 14, final paragraph, there are more recent papers than ref. 47 on the relationship between ESCRT proteins, lysosome acidification and canonical Wnt signaling (Tejeda-Munoz et al PNAS 2019; Albrecht et al. Cell Rep. 2020).

In sum, this high-quality work reveals novel molecular mechanisms that drive ER+ breast cancer. It is an original discovery and will be of interest to a wide readership.

REVIEWER COMMENTS

Reviewer #1 (Remarks to the Author):

In this manuscript, Rodgers et al. report that INPP4B exerts an oncogenic function in promoting ER+ breast cancer growth through enhancing late endosome formation/Wnt-b-catenin signaling activation. Through data mining the authors observed that increased INPP4B mRNA and protein levels in ER+ breast cancer patients. To examine INPP4B function in regulating ER+ breast cancer growth, the authors overexpressed INPP4B in ER+ breast cancer lines MCF7 and T47D. As a result, overexpression of INPP4B facilitated cell growth albeit reducing AKT activation. Further the authors showed that INPP4B expression facilitated PI(3)P accumulation and late endosome formation, and blocking late endosome function by Hrs depletion partially rescued INPP4B overexpression induced growth advantages. Moreover, the authors found INPP4B facilitated Wnt/b-catenin activation which is also necessary for INPP4B expression induced growth increase. Overall, the authors reveal an interesting oncogenic function for INPP4B in PIK3CA mutated ER+ breast cancer through controlling PI(3)P/late endosome/Wnt signaling. The manuscript is well written and experiments were designed in a rational manner to support major conclusions. However, this study heavily relies on data obtained from INPP4B overexpression in an ER+ breast cancer line already with high INPP4B expression that may be artificial. In addition, a lack of causal relationship for the proposed signaling reduces the enthusiasm from the reviewer.

Major concerns:

1. Instead of relying on INPP4B overexpressing MCF-7 and T47D lines, the authors need to obtain more evidence from endogenous INPP4B depleted MCF-7 or T47D or other ER+ breast cancer lines.

Response: We acknowledge the reviewers concerns but there is a complexity with respect to INPP4B which paradoxically acts as an oncogene when overexpressed in ER+ breast cancer cells and a tumour suppressor when depleted. This relates to the relative ratios of PI(3,4)P₂, the substrate of INPP4B that can activate oncogenic AKT signalling, *versus* PI(3)P, the product of INPP4B PI(3,4)P₂ hydrolysis that we show here can activate oncogenic Wnt signalling. INPP4B depletion increases PI(3,4)P₂ levels at the plasma membrane and early endosomes leading to enhanced AKT activation (PMID: 30174291). Consistent with this, we and others previously reported that INPP4B depletion enhanced MCF-7 cell proliferation and xenograft tumor growth via AKT hyperactivation (PMID: 21127264, 26411369). In other tumours INPP4B overexpression has been shown to be oncogenic, although the molecular basis of this effect is still emerging. As we demonstrate in this manuscript, a subset of primary ER+ breast cancers exhibit increased INPP4B expression (20-46%) (**Figure 1**), and MCF-7 and T47D cells exhibit an intermediate level of INPP4B expression (PMID: 31068700), therefore we propose that overexpression rather than depletion of INPP4B in these cells is the best experimental model to reflect these pathological findings. In this context, we show INPP4B overexpression suppressed AKT signalling but enhanced cell proliferation via an alternate oncogenic pathway through late endosome formation and Wnt signaling. We have amended the discussion to clarify these issues (lines 431-437). In addition, we have also provided further experimental data to demonstrate the effects of INPP4B overexpression are dependent on its PI(3,4)P₂ 4-phosphatase activity and not due to artificial overexpression effects. Expression of PI(3,4)P₂ 4-phosphatase-dead INPP4B did not recapitulate the enhanced late endosome formation (**new Supplementary Fig. 4d and 4e**), Wnt activation (**new Supplementary Fig. 7b**) or cell proliferation (**Fig. 2i-p, Supplementary Fig. 2c and 2d**) observed in wild-type INPP4B overexpressing cells. We also demonstrated that INPP4B shRNA depletion

reduced PI(3)P levels on late endosomes (**Supplementary Fig. 3i**) and late endosome formation (**Supplementary Fig. 4a-c**), providing further confirmation of the signaling mechanisms mediated by INPP4B in our proposed model.

2. The causal relationship for the proposed INPP4B/PI(3)P/late endosome formation/Wnt signaling would need to be established. For example, how PI(3)P/late endosome regulates Wnt signaling?

Response: Our data is consistent with the interpretation that INPP4B increases PI(3)P levels on late endosomes, which enhances Hrs-dependent late endosome formation to promote the trafficking of GSK3 β to late endosomes and its lysosomal degradation, leading to Wnt/ β -catenin hyperactivation. The causal relationship was established by depleting the PI(3)P-binding protein, Hrs, in INPP4B-overexpressing cells, which rescued the increase in late endosome formation (**Fig. 5a**), GSK3 β degradation (**Fig. 6f and 6g**), colony size in soft agar (**Fig 5c and 5d**) and tumor growth (**Fig. 5e-g**). To further strengthen our proposed mechanism, we have now shown that wild-type but not PI(3,4)P₂ phosphatase-dead INPP4B promotes increased late endosome numbers (**new Supplementary Fig. 4d and 4e**) and AXIN2 mRNA levels (**new Supplementary Fig. 7b**), suggesting that PI(3,4)P₂ to PI(3)P conversion by INPP4B is essential for late endosome formation and Wnt activation. Furthermore, we have now shown that INPP4B promotes GSK3 sequestration into late endosomes/lysosomes (**new Fig 6h and 6i**), which demonstrates a clearer causal link between late endosomal PI(3)P and Wnt signaling.

3. This study focuses on PIK3CA mutated ER+ breast cancer settings, but some key questions are not revealed, including (1) Is ER necessary to prime INPP4B function in this setting? (2) Is PIK3CA oncogenic mutant necessary to prime INPP4B function in this setting?

Response: We agree that this is an important question that was not completely addressed in the manuscript. Therefore, we have further examined INPP4B function in MCF-10A mammary epithelial cells, which are ER-negative and express wild-type PIK3CA, in order to assess the dependence of INPP4B on ER-positivity and PIK3CA mutation. Altogether, our data shows that INPP4B overexpression in MCF-10A increased the number of late endosomes (**new Supplementary Fig. 4d and 4e**), AXIN2 mRNA levels (**new Supplementary Fig 7b**) and enhanced 2D and 3D cell proliferation (**Fig 2i-p**). We have therefore added to the discussion that “*Interestingly, INPP4B overexpression also enhanced Wnt/ β -catenin activation and proliferation of MCF-10A cells, which are ER⁻ and express wild-type PI3Ka, suggesting that ER-positivity or mutant-PIK3CA are dispensable for INPP4B oncogenic signaling although these factors could enhance signaling.*” (lines 437-441).

Minor concerns:

1. Figure 1: INPP4B DNA alteration should also be examined using these datasets, especially in ER+ group;

Response: Examination of the METABRIC and TCGA datasets revealed only 1% of cases have INPP4B genetic alterations (mutations, truncations, amplifications or deletions), which we have now reported in our manuscript (lines 117-119).

2. Line 136: should be Figure 2D and 2E.

Response: This typographical error has been amended (now line 138)

3. Figure 2D/2E: growth between normal serum condition and serum deprivation condition should be compared.

Response: We have performed this experiment under normal serum conditions as requested, showing that >95% of GFP-vector and GFP-INPP4B cells are proliferating under these conditions (**new Supplementary Fig. 2f**).

4. Figure 2F: AKT- if this is total AKT or AKT1 should be clarified. Reduction of AKT phosphorylation upon ectopic INPP4B expression is weak. INPP4B/GFP blot is needed in this panel to reveal INPP4B expression levels.

Response: We have changed “AKT” to “AKT(pan)” in the figure (**Fig. 2f**) to clarify that total AKT protein was examined and not AKT1. We agree that AKT phosphorylation levels were modestly reduced and have now indicated this in the results (line 140). We have also included an immunoblot demonstrating INPP4B overexpression for this experiment (**Fig. 2f**).

5. Figure 2G-2H: normalization should be done using time 0 (starvation) condition as the basal line to reveal dynamic AKT phosphorylation changes upon EGF stimulation. In addition, a catalytic dead INPP4B is needed here as an additional negative control.

Response: We agree with the reviewer that this method of normalization better informs about the dynamic changes to phospho-AKT levels in response to EGF, and we have included this data (**new Fig. 2g and 2h**). However, because there are very high phospho-AKT levels following 5 minutes of EGF stimulation, this normalization method makes it difficult to statistically compare differences between GFP-vector and GFP-INPP4B cells at timepoints where phospho-AKT levels are very low, such as unstimulated conditions or at later timepoints. Thus, we have also normalised to the control cells at each time point, a method which we have previously reported (PMID: 21127264). We have also included the requested additional negative control showing that wild-type but not phosphatase-dead INPP4B suppresses EGF-stimulated AKT activation (**new Supplementary Fig. 2p and 2q**).

6. Figure 2 is largely depending on overexpression of INPP4B- given as the authors claimed earlier that increased INPP4B mRNA/protein expression is observed in ER+ breast cancer, what is the rationale here to rely on further INPP4B overexpression for these studies? Why not just examine what happens upon knockdown of endogenous INPP4B?

Response: See major concerns, point 1. Examination of *INPP4B* mRNA levels in ER+ cell lines using the Cancer Cell Line Encyclopedia showed that MCF-7 and T47D cells have intermediate *INPP4B* expression compared to a number of other ER+ cell lines (PMID: 31068700). As INPP4B expression is increased in a subset of ER+ breast cancers (20-46%), we chose to overexpress rather than deplete INPP4B to model these pathological findings.

7. Figure S2F: Sample loadings are not even.

Response: We have added densitometry quantification to demonstrate more clearly that INPP4B does not significantly affect phospho-SGK3 levels (**new Supplementary Fig. 2j**).

8. Figure S2G: An increased PTEN protein level is observed in GFP-INPP4B sample- this is not consistent with the authors' claim.

Response: We acknowledge that PTEN protein levels appeared to be higher in the GFP-INPP4B sample due to higher sample loading. We have repeated this experiment multiple times and densitometric analysis confirmed that INPP4B does not significantly affect PTEN protein levels (**new Supplementary Fig. 2g and 2h**).

9. Supple Figure 2B: what is the band between EV and INPP4BWT lane?

Response: Empty lanes were left between each loaded sample in this immunoblot, and the band is a marginal amount of spill-over between samples 2 (empty vector + doxycycline) and 3 (Myc-INPP4B^{WT} – doxycycline).

10. Figure 2L: Ki67 should be normalized to DAPI stain. In contrary to Figure 2I, no obvious increased in the size of acini, nor increased number of DAPI stain is observed in Figure 2L upon expression of WT-INPP4B.

Response: The axis label of this figure has now been amended to “Ki67-positive nuclei” to clarify that this analysis was normalised to the DAPI nuclear stain (**Fig. 2j**). We have also included quantification demonstrating that there is no significant increase in the size of INPP4B-overexpressing acini size after only 7 days in culture (**new Fig. 2k**) when acini are still small and immature, but an increase in size was evident after 14 days when acini had reached their terminal size (**Fig 2m**).

11. Figure 3B: for IP-mass spec experiments, the rationale for starvation/EGF stimulation would need to be provided.

Response: We have amended the results to indicate that EGF stimulation was used to activate class I PI3K signalling (lines 196-198).

12. Figure 3C: An interaction between INPP4B and Rab7A would need to be confirmed at endogenous levels.

Response: We attempted to co-immunoprecipitate endogenous INPP4B and Rab7 proteins but these experiments were ultimately unsuccessful. Co-immunoprecipitation of two endogenous proteins is not always possible, as the antibody epitope may be buried within the complex, the antibody could interfere with complex formation or the interaction may be transient and difficult to capture endogenously. However, we have shown that endogenous Rab7 can bind GFP-INPP4B (**Fig. 3c**), and that endogenous INPP4B can bind HA-Rab7 (**Supplementary Fig. 3a**), demonstrating that both endogenous INPP4B and Rab7 can participate in this complex.

13. Figure S3A: the input signals are not even, thus the interpretation of binding differences is not reliable.

Response: Protein loading in this figure is similar between samples as indicated by GAPDH, but we acknowledge that inactive Rab7(T22N) does not express as highly as wild-type Rab7 or active Rab7(Q67L) (**Supplementary Fig. 3a**). This difference in expression is consistent with other studies (PMID: 28860274). We have used this as a qualitative rather than quantitative assessment to show that INPP4B is able to bind Rab7 WT, Q67L and T22N, but we agree that interpretation of any differences in binding affinity is not possible from this analysis and have rephrased this sentence in the revised results as follows: “*Immunoprecipitation of HA-Rab7^{WT}, HA-Rab7^{Q67L} (constitutively GTP-bound) or HA-Rab7^{T22N} (constitutively GDP-bound), showed that endogenous INPP4B is able to bind both GTP and GDP-loaded Rab7*” (lines 204-206).

14. Figure 4: the authors nicely showed that overexpression of INPP4B facilitates late endosome formation. Is this accompanied by increased levels of cellular PI(3)P? Does blocking PI(3)P prevents INPP4B overexpression induced late endosome formation?

Response: Our data shows that INPP4B overexpression increases total intracellular PI(3)P levels (**Fig. 3f**) and increased PI(3)P on late endosomes (**Fig. 3i**) using the 2xFYVE/Hrs recombinant biosensor, which is a standard method used to measure intracellular PI(3)P levels (PMID: 20736309). INPP4B depletion conversely reduced PI(3)P levels on late endosomes, and late endosome formation (**Supplementary Fig. 3i**). We have included new data showing that wild-type but not PI(3,4)P₂ phosphatase-dead INPP4B increases late endosome numbers (**new Supplementary Fig 4d and 4d**), demonstrating that PI(3,4)P₂ to PI(3)P conversion is required for late endosome formation. There are no mammalian 4-kinases that convert PI(3)P back to PI(3,4)P₂, thus it is not possible to reverse INPP4B-mediated PI(3)P synthesis.

15. Figure S5C/S5D: genetic depletion of PIK3CA is needed to validate inhibitor effects on late endosome formation.

Response: As requested, we have performed *PIK3CA* depletion using two distinct siRNAs (**new Supplementary Fig. 5g and 5h**), which also suppressed the increased late endosome formation in INPP4B-overexpressing cells similar to BKM120 and BYL719 PI3K inhibitors (**new Supplementary Fig. 5i and 5j**).

16. Figure 5A: shHrs reduces CD63+ late endosome- but it seems this late endosome reduction does not affect 3D growth or tumor growth. Figure 5 reveals a critical function for Hrs in mediating INPP4B overexpression induced growth advantage, but if PI(3)P or late endosome formation plays a role here, or if this is mediated by Hrs function on late endosome is not well established.

Response: Our current data would suggest that in ER+ breast cancer, Hrs may not affect 3D growth or tumor growth in the absence of INPP4B overexpression. The xenograft tumor growth experiments were terminated when INPP4B-overexpressing tumors reached the ethical endpoint (1 cm³ tumor volume). At this timepoint, control and Hrs knockdown tumors were likely too small (mean 147 cm³ and 135 cm³ respectively) to observe any effects of Hrs depletion alone (**Fig. 5e-g**). This may also be true for anchorage-independent cell growth assays, where control and Hrs knockdown colonies are likely too small to observe any differences (**Fig. 5c and 5d**). Therefore, we cannot exclude that Hrs depletion affects 3D or tumor growth as these experiments were not optimised to appropriately

answer this question, which is beyond the scope of our study but should be investigated in future studies.

17. Figure 6D: increases of non-phospho- β -catenin are quite minor in Wnt3a treated samples. In addition, in GFP-INPP4B expressing cells upon double treatment there is no further reduction of GSK3 β but why there is a further increase in non-phospho- β -catenin signal? Does this suggest this change in non-phospho- β -catenin is not mediated by GSK3 β protein level changes?

Response: Measuring expression of the Wnt-specific target gene, *AXIN2*, is a commonly used readout of Wnt/ β -catenin signaling (PMID: 11809808, 11940574), and we found this to be the most reliable method to detect Wnt activation in our breast cancer models (**Fig. 6b and 6c, Supplementary Fig. 7a and 7b**). Although β -catenin and GSK3 β are critical Wnt/ β -catenin signaling proteins, both have Wnt-independent functions and thus may not always strictly correlate with Wnt activation. For example, β -catenin complexes with E-cadherin to regulate cell junctions (PMID: 28412244), whereas the majority of GSK3 β exists independent of the β -catenin destruction complex as an effector of AKT signaling (PMID: 25435019, 10913153, 19850932). We acknowledge that GSK3 β and active- β -catenin levels do not always correlate in our experiments and can only speculate on the reasons for this. We found Wnt stimulation alone did not affect total GSK3 β protein levels (**Fig. 6d**), as reported by previous studies (PMID: 21183076), despite its known recruitment to late endosomes/lysosomes under these conditions. The reasons for this are not clear but may be a consequence of GSK3 β transcriptional upregulation or delayed lysosomal degradation (PMID: 21183076). However, INPP4B-mediated increased late endosome formation decreased GSK3 β levels under normal and Wnt-stimulated conditions (**Fig. 6d**), suggesting that in cells with INPP4B overexpression, GSK3 β degradation is enhanced and may occur more rapidly than its transcriptional replenishment. Further studies would be needed to dissect the relationship between active- β -catenin and GSK3 β levels and Wnt target gene transcription.

18. Figure 6I: can effects of pharmacological inhibitors be decapitated by genetic manipulation of β -catenin signaling?

Response: To rescue the increased proliferation of GFP-INPP4B cells, we chose to use inhibitors rather than genetic downregulation of Wnt/ β -catenin signaling as this allowed us to titrate the inhibitor dose to a concentration that had little effect on GFP-vector cell proliferation. As the reviewer requested, we attempted to rescue the phenotype via genetic manipulation of Wnt/ β -catenin signaling. As β -catenin has Wnt-independent effects such as regulation of E-cadherin junctions (PMID: 28412244), we chose to target the TCF transcription factors that promote Wnt target transcription in response to β -catenin binding. We expressed dominant negative mutants of TCF4 (Addgene #19284) or TCF4E (Addgene #32739) that lack the β -catenin binding motif, which is a commonly used method to suppress Wnt/ β -catenin signaling (PMID: 22157225). However, dominant negative TCF4 or TCF4E expression severely decreased both GFP-vector and GFP-INPP4B cell proliferation under serum-free conditions. This is consistent with previous findings that expression of dominant negative TCF4 inhibits MCF-7 cell proliferation (PMID: 20139903). Thus we were unable to achieve a level of Wnt inhibition that had little effect on the GFP-vector control cells in order to perform these rescue experiments. As Wnt/ β -catenin rescue experiments were performed using three independent validated inhibitors, IWP-2 (PMID: 27698112, 32311594), LGK-974 (PMID: 32848220, PMID: 31367040) and IWR-1-endo (PMID: 31511693, 33360774), which target multiple

components of the pathway, we are confident that the effects of these inhibitors are specific for the Wnt/ β -catenin pathway.

Reviewer #2 (Remarks to the Author):

The authors report on how a phosphatase the converts PI(3,4)P₂ into PI(3)P, INPP4B, contributes to breast cancer through membrane trafficking that causes increased Wnt signaling. The authors provide a compelling and thorough analysis of INPP4B from the level of cell signaling to in vivo using mouse models. This discovery is important as it offers novel insights into the field of cancer cell signaling downstream of PI3K and PTEN. The new findings on the oncogenic activity of INPP4B offer a mechanism to understand how INPP4B contributes to the progression of ER+ breast cancer. Previous work demonstrated that INPP4B was inactivated in triple negative breast cancer. Conversely, overexpression of INPP4B had been reported to increase tumorigenesis on other cancers. Here, the authors report that in the case on ER+ breast cancers, expression of INPP4B is increased at the level of both mRNA and protein (pg. 5). Authors go on to show that INPP4B promotes proliferation and growth, even though AKT signaling is inhibited (pg. 6). Mechanistically, INPP4B interacts with Rab7 at late endosomes and subsequently increases endosomal PI3K conversion of PI(3,4)P₂ to PI(3)P.

Main Comments

1) The conclusions of the work are nicely summarized in a schematic depiction that is currently presented as the last figure of the supplemental text. It is essential that this model be included as part of the final figure in the main text. It tells the entire tale and will greatly help the readership. Other data can be accommodated in the supplemental material to make space for this explanatory diagram.

Response: We agree that this schematic should be presented in the main manuscript and have moved it as requested (**now Fig. 7e**).

2) Given the recent emerging roles of lysosomal catabolism and Wnt-induced macropinocytosis in promoting cancer cell metabolism, the authors could include discussion about the possible physiological implications of lysosomal activity in ER+ breast cancer in relation to other cancer types. They convincingly show that INPP4B activates the canonical Wnt pathway through increased lysosomal degradation of GSK3, increases total number of lysosomes, and endocytosis and degradation of extracellular serum protein using BSA-DQ. Authors should therefore discuss the general implications of late endosome formation on cell signaling in cancer.

Response: We thank the reviewer for this useful suggestion, and have included the following in the discussion (lines 499-504): “*Recent studies have also found that Wnt signaling increases endolysosomes and macropinocytosis to promote protein uptake and degradation(48, 50). Macropinocytosis enhances the proliferation and drug resistance of breast cancer cells(51), thus INPP4B regulation of Wnt signaling and lysosome activity could also affect cancer cell metabolism and should be investigated in future studies.*”

Minor Comments

1) The authors clearly demonstrate that the oncogenic roles of INPP4B occur upstream of Wnt signaling through endolysosomal signaling, as Wnt inhibition or HRS knock down was sufficient to block tumor growth. HRS depletion studies were performed in cultured cells, agar colony assay, and in xenograft mice models. They show that GSK3 is degraded in lysosomes on page 11 (Figure

6 and Supplementary Figure 7) by protein levels in the presence of GFP-INPP4B expression, but not at the level of mRNA. Further, GSK3 levels were rescued with HRS knockdown. This supports a model whereby GSK3 is being degraded in lysosomes versus proteasomes. We suggest that including the ratios of GSK3 protein expression relative to GAPDH under the western blot lanes of Figure 6D could help readers better appreciate the decrease in GSK3 protein levels.

Response: As the reviewer suggested, we have included quantification of GSK3 β protein levels (**new Fig. 6e**).

2) Figure 6G makes the very important point that INPP4B increases sequestration of GSK3 in late endosomes. While the data as is makes the point, it is not very impressive. Digitonin usually gives better results than saponin. The authors could consider either co-staining with a late endosomal marker such as CD63, or the more exacting protease protection in live cells described in Albrecht et al PNAS 2018. Figure 6G should be improved.

Response: We agree that the data showing enhanced GSK3 β sequestration into late endosomes could be improved and attempted the experiments suggested by the reviewer. We undertook experiments using several digitonin permeabilization conditions, and found these to be much less effective at clearing cytoplasmic proteins in MCF-7 cells than saponin treatment. Unfortunately, the CD63 antibodies (cat # H5C6) are the same species and isotype (mouse IgG1) as the GSK3 β antibodies (cat # 610201) thus we were unable to perform co-labelling experiments. We also attempted the live cell proteinase protection assay as described in Albrecht et al. 2018 (PMID: 29773710), however, proteinase K treatment (1 μ g/mL) induced MCF-7 cell shrinkage indicating apoptosis, and the proteinase K + Triton X-100 (0.1% v/v) control resulted in significant cell death. Instead, we performed a protease protection assay and extracted proteins for immunoblot analysis as previously described (PMID: 19265192, PMID: 29773710). We found GSK3 β was protected from proteolysis in GFP-INPP4B cells but not GFP-vector cells (**new Fig. 6h**). In agreement with this, RFP-GSK3 colocalised with the late endosome/lysosome marker, LysoTracker, in GFP-INPP4B but not GFP-vector cells (**new Fig. 6i**). This new data supports our contention that INPP4B promotes GSK3 β sequestration into late endosomes, leading to its lysosomal degradation.

3) On page 14, final paragraph, there are more recent papers than ref. 47 on the relationship between ESCRT proteins, lysosome acidification and canonical Wnt signaling (Tejeda-Munoz et al PNAS 2019; Albrecht et al. Cell Rep. 2020).

Response: We have amended the discussion to include these additional references.

In sum, this high-quality work reveals novel molecular mechanisms that drive ER+ breast cancer. It is an original discovery and will be of interest to a wide readership.

REVIEWERS' COMMENTS

Reviewer #1 (Remarks to the Author):

The authors have addressed previous concerns and now I agree it is suitable for publication in Nature Communications.

Reviewer #2 (Remarks to the Author):

This is a very important paper showing that a phosphatase that converts PIP2 into PI(3)P can activate canonical Wnt signaling in ER+ breast cancer. The authors addressed all my comments and I am happy to recommend this paper be published as is. The authors discovered a very important signaling cross-talk in cancer, congratulations.